

# Characterisation of biofluorescent aerosol emissions over winter and summer periods in the United Kingdom

Elizabeth Forde[1], Martin Gallagher[1], Virginia Foot[2], Roland Sarda-Esteve[3], Ian Crawford[1], Paul Kaye[4], Warren Stanley[4], and David Topping[1]

[1]Centre for Atmospheric Science, School of Earth and Environmental Science, University of Manchester, Manchester, U.K.
[2]Defence Science and Technology Laboratory, Porton Down, Salisbury, U.K.
[3]Laboratoire des sciences du climat et de l'environnement, Saclay, France
[4]Particle Instruments Research Group, University of Hertfordshire, U.K.

*Correspondence to:* Elizabeth Forde (elizabeth.forde@manchester.ac.uk)

**Abstract.** Primary biological aerosol particles (PBAP) are an abundant subset of atmospheric aerosol particles which comprise viruses, bacteria, fungal spores, pollen, and fragments such as plant and animal debris. The abundance and diversity of these particles remain poorly constrained, causing significant uncertainties for modelling scenarios and for understanding the potential implications of these particles in different environments. PBAP concentrations were studied at four different sites in

the United Kingdom (Weybourne, Davidstow, Capel Dewi, and Chilbolton) using an ultra-violet light induced fluorescence (UV-LIF) instrument, the Wideband Integrated Bioaerosol Spectrometer (WIBS), versions 3 and 4.

Using hierarchical agglomerative cluster (HAC) analysis, particles were statistically discriminated between. Fluorescent particles and clusters were then analysed by assessing their diurnal variation and their relationship to the meteorological variables, temperature and relative humidity, and wind speed and direction. Using local land cover types, sources of the suspected

fluorescent particles and clusters were then identified.

Most sites exhibited a wet discharged fungal spore dominance, with the exception of one site, Davidstow, which had higher concentrations of bacteria, suggested to result from the presence of a local dairy factory. Differences were identified as to the sources of wet discharged fungal spores, with particles originating from arable and horticultural land at Chilbolton, and improved grassland areas at Weybourne. Total fluorescent particles at Capel Dewi were inferred to comprise two sources, with

bacteria originating from the broadleaf and coniferous woodland and wet discharged fungal spores from nearby improved grassland areas, similar to Weybourne.

The use of HAC and a higher fluorescence threshold (9SD) produced clusters which were considered to be biological following the complete analysis. More knowledge of the reaction of speciated biological particles to differences in meteorology, such as relative humidity and temperature would aid characterisation studies such as this.

*Copyright statement.*





## 1 Introduction

Primary biological aerosol particles (PBAP), or simply, bioaerosols, are a subset of atmospheric aerosol particles produced from a range of sources within the biosphere. The constituents of PBAP vary in size and abundance, and include viruses (0.01 - 0.3 μm), bacteria and associated agglomerates (0.1 - 10μm), fungal spores (1 - 30μm), pollen (5 - 100μm), and fragments

such as plant and animal debris (Després et al., 2012). Biological particle dispersal has implications for agricultural, animal, and human health (Polymenakou et al., 2008; Fisher et al., 2012; D'Amato et al., 2002; Douwes et al., 2003), whilst also influencing the hydrological cycle and climate, by acting as ice nuclei and cloud condensation nuclei (Pöschl et al., 2010; Schumacher et al., 2013; Huffman et al., 2013; Hoose and Möhler, 2012; Pratt et al., 2009; Pope, 2010). However, the extent of these impacts is highly uncertain, in part, owing to difficulties in characterising the identity and abundance of different

biological particulates in the atmosphere.

### 1.1 Influences on PBAP number

The abundance and presence of PBAP in different regions is impacted by local scale meteorology, including, but not limited to, temperature and relative humidity (Jones and Harrison, 2004). Previous studies have identified relationships between temperature and relative humidity in relation to two abundant fungal spores, Alternaria and Cladosporium. These two fungal spores

strongly correspond with air temperature, but have negative correlations with relative humidity (Oliveira et al., 2009; Corden and Millington, 2001; Fernández-Rodríguez et al., 2017; Grinn-Gofroń and Mika, 2008), for example, Cladosporium has been found to be more abundant at temperatures ranging from 20°C to 24°C depending on the species (Fernández-Rodríguez et al., 2017).

Alternaria and Cladosporium are examples of dry spore discharged fungi, in which spores are dislodged by air currents or

other forces, as opposed to liquid jets or droplets in the air (Elbert et al., 2006; Burch and Levetin, 2002), and are produced when temperatures are high and relative humidity is low (Crandall and Gilbert, 2017; Elbert et al., 2006). In comparison, actively wet discharged fungal spores such as Ascospores and Basidiospores, are found in the air during cooler nighttime and evening hours, and during wet or humid conditions such as in the early morning when there is an increase in relative humidity (Burch and Levetin, 2002; Crandall and Gilbert, 2017; Elbert et al., 2006; Oliveira et al., 2009; Gabey et al., 2010).

The influence of temperature on both fungal spores and bacteria between differing soil types illustrated that the optimum temperatures for bacteria growth are higher compared to fungal spores (Pietikäinen et al., 2005), with bacterial production found to correlate with increases in temperature, and in a salt-marsh estuary occurring when temperatures were >22°C (Apple et al., 2006). As such, bacteria concentrations have been found to be highest in summer and autumn, and lower in spring and winter (Fang et al., 2007; Després et al., 2012), which has been suggested to result from frozen ground, and lack of foliage

(Bowers et al., 2011).

The diurnal patterns of biological particles differ and for bacteria these particles have been found to increase at sunrise, decrease during solar noon hours, gradually increase until sunset, then decrease into the evening, with lowest concentrations between 21:00 - 05:00 (Shaffer and Lighthart, 1997). A diurnal cycle has also been noted for pollen particles, which have



shown increases in concentration near sunrise, with a peak a few hours later, prior to a gradual decline in the afternoon, and minor concentrations during the night (Ogden et al., 1969).

Temperature has also been found to be the main driver for controlling pollen release, and for some species it is temperature, alongside precipitation, that controls the amount of pollen that is produced (Duhl et al., 2013). Similarly, direct rainfall events

have been found to show a stronger relationship with bacteria concentrations, as identified in a study conducted in a Colorado forest (Crawford et al., 2014). Additionally, using an Aerodyne aerosol mass spectrometer in Switzerland, rainfall was correlated with bacteria like particles at one site, and compared to the total campaign average these concentrations increased by ~24 % (including 30 min after the precipitation event), as a result of particle resuspension following impaction with the ground and other surfaces (Wolf et al., 2017).

Increases in PBAP concentrations are not only related to meteorological conditions, and instead the proximity of the site to different land cover types, alongside wind speed and wind direction data, is to be considered in order to identify distinctive emission patterns and factors. It has been found that in arable and agricultural areas, increases in bioaerosol concentration may result from the maturing of crops and tree foliage (Grinn-Gofroń and Mika, 2008), or combine harvesting and grass mowing (Corden and Millington, 2001). For example, in Denmark agricultural areas were found to be the main source of airborne

Alternaria, the sources of such originating locally or regionally, with some intermittent long distance transport (Skjøth et al., 2012). The nature of the surrounding area can be identified using land cover maps, and the use of such have been applied to analyse the distribution of pollen vegetation in the United Kingdom (McInnes et al., 2017), and combined with remote sensing to create a pollen inventory in Aarhus, Denmark (Skjøth et al., 2013). Previous studies have identified the relationship between different sites and bacterial particle concentrations, with lower concentrations found at coastal and rural sites, and

higher concentrations at forest and urban sites (Shaffer and Lighthart, 1997; Harrison et al., 2005). However, the influence of wind speed and wind direction on airborne biological particle concentrations is unclear, and at two sites in Switzerland no linear correlation coefficient was found between bacteria and wind speed and wind direction (Wolf et al., 2017). Whilst, in comparison, in a study focussing on the influence of different meteorological factors, wind speed was found to have the most pronounced influence on bacterial concentrations (Mouli et al., 2005).

**1.2 UV-LIF discrimination**

Biological particles fluoresce when illuminated with ultra-violet light, owing to the intrinsic presence of bio-fluorophores, such as nicotinamide adenine dinucleotide phosphate (NAD(P)H), tryptophan, and riboflavin, which auto-fluoresce when excited by UV radiation. Ultra-violet light induced fluorescence (UV-LIF) instruments work on this principal, with the detection channels in these instruments measuring fluorescence coinciding with the maximum emission spectrum of each biological fluorophore

(Kaye et al., 2005). Tryptophan, an amino acid, is excited at ~280nm and emits fluorescence from 300-400nm, the co-enzyme, NAD(P)H, is excited between 270nm and 400nm and emits between 400 - 600nm (Kaye et al., 2005), and Riboflavin is mainly excited at ~450nm and emits at around 520 - 565nm (Hill et al., 2009; Lakowicz, 2006).

UV-LIF instrumentation allows for real-time measurements, providing instantaneous data without the need for constant maintenance. This allows for continuous monitoring for extended periods of time as opposed to traditional sampling techniques



which although allow for accurate identification of particle type and species, are often laborious, with poor time resolution, and may suffer from potential identification biases when manually counting particles (Spracklen and Heald, 2014; Robinson et al., 2013).

There is a potential interference risk from non-biological fluorescent particles which can possess similar fluorescence prop-
erties to PBAP, and can fluorescence at wavelengths used by UV-LIF instruments. Chemical pollutants from vehicles, such as diesel particulates and other secondary organic aerosols (SOA) are known to fluoresce upon excitation wavelengths, especially in sub micron ranges (O'Connor et al., 2014; Perring et al., 2014). This has been experienced in polluted environments, with combustion-type particles found to dominate the 1 - 2µm size range (Yu et al., 2016). The most common interferents include Polycyclic Aromatic Hydrocarbons (PAHs) and SOA, with those less likely to cause interference comprising humic
like substances (HULIS), mineral dust, and soot due to weak signal intensities (Pöhlker et al., 2012). However, due to the weak intensities of these interferents overall, most ambient fluorescent particles are likely to be dominated by biological particles (Pöhlker et al., 2012).

During atmospheric transport, airborne particles are subject to aging, chemical and physical transformations, and fragmentation affecting the fluorescent characteristics of the particle (Fröhlich-Nowoisky et al., 2016; Pöschl and Shiraiwa, 2015).
Biological particles have also been found to accompany the transport of dust within the atmosphere, and previous studies have found that dust events have resulted in a ten-fold increase in airborne micro-organisms, such as fungal spores and pathogens (McCarthy, 2001; Polymenakou et al., 2008). Collectively, these processes can make comparisons between ambient and laboratory sampling of the same particle difficult.

### 1.3 Hierarchical Agglomerative Cluster (HAC) Analysis

The use of hierarchical agglomerative cluster analysis (HAC) to distinguish and statistically segregate different types of biological particles is required as UV-LIF instruments do not provide information on particle genus or species (described further in Section 2.3). The use of this method has been applied for analysis of data from a Colorado pine forest (Robinson et al., 2013; Crawford et al., 2014; Huffman et al., 2013; Gosselin et al., 2016), a high altitude site in central France (Gabey et al., 2013), and the Brunt ice shelf in Antarctica (Crawford et al., 2017). This has also been applied to laboratory data and has been shown
to effectively segregate between Polystyrene Latex Spheres (PSLs) of different sizes and doping (Crawford et al., 2015).

### 1.4 Scope

Recent work, and ongoing areas of research, has included emission modelling for pollen particles as based on observed pollen counts within the United States (Wozniak and Steiner, 2017), and others assessing the impacts of bioaerosols on human health, focussing specifically on the ability of these particles to produce damaging oxidative reactions in human lungs (Samake et al.,
2017). This study reports the analysis of measurements taken at four different sites within the United Kingdom, during different times of year, using an ultra-violet light induced fluorescence (UV-LIF) instrument, the Wideband Integrated Bioaerosol Spectrometer (WIBS). Using a HAC approach, different clusters are statistically discriminated between. The resulting fluorescent and cluster concentrations between sites are analysed in relation to meteorological conditions, focusing specifically on temper-



ature and relative humidity, and in relation to wind speed and direction. HAC cluster solutions and responses to meteorological drivers from each site are compared with local land cover type to identify distinctive emission patterns and factors. This is the first comparison study of measurements taken from four different sites in the United Kingdom using a UV-LIF instrument, and attempts to classify particles from these sites using HAC, meteorological data, and land cover mapping, collectively, and infer

emission types in association with different land cover types. Contrary to previous work, this is additionally the first use of a differing fluorescent threshold of 9 standard deviations (SD) compared to traditionally 3SD, in an ambient setting.

## 2 Methods

### 2.1 Site Descriptions

Bioaerosol measurements were conducted at four different sites within the United Kingdom during different years, and time

of year (Table 1). These sites were Chilbolton in Hampshire, Davidstow in North Cornwall, Weybourne on the coast of north Norfolk, and Capel Dewi near Aberystwyth, Wales (Fig. 1). At each site, the instruments were connected to a PM10 inlet, with the height of this connection varying between sites.

To identify the land cover characteristics of each site, the Centre for Ecology and Hydrology Land Cover Map 2015 (LCM2015) was used to provide background information per site (Rowland et al., 2017). The use of a previous version of

the LCM2015 was used by McInnes et al. (2017) to produce location maps of trees, weeds, and grasses that are associated with allergies and asthma. Whilst Skjøth et al. (2012) utilised the Corine Land Cover 2000 dataset to identify agricultural areas under rotation and in harvest in relation to Alternaria spore concentrations in Denmark. Here, using the LCM2015 at each site, a land cover class which is common is 'Improved Grassland', which is distinguished from semi-natural grasslands owing to its higher productivity and lack of winter senescence, whilst those defined as 'Arable and Horticulture' comprise annual crops,

perennial crops (e.g. orchards), and freshly ploughed land.

Chilbolton is situated within Hampshire on the southern coast of England, and data collection was conducted at Chilbolton Observatory situated at the edge of Chilbolton village. Measurements were conducted from the 20 January to the 20 March 2009, with the instrument connected to the PM10 inlet at a height of 8m. The area surrounding Chilbolton Observatory comprises mainly arable and horticultural land, with some broadleaf woodland and improved grassland covered areas situated

around the site. To the south-east of the observatory there is an industrial composting facility and mushroom farm.

Data collection from Davidstow, in North Cornwall, was conducted from the 25 June to the 28 August 2013, in which the instrument was connected to a 10m sampling line. The land cover around the ground site comprises predominantly improved grassland coverage, with some urban and suburban land cover, arable and horticultural land, and heather covered areas. The Davidstow Airfield Runway is located south-easterly from the ground site, in a north-westerly direction there resides a dairy

factory, the largest producers of cheddar cheese in Britain.

Capel Dewi is situated near Aberystwyth in west Wales, and from the 18 February to 3 June 2013 data were collected from the NERC MST Radar Site, at an inlet height of 3m. The location of the site is within, and mostly surrounded by, an improved





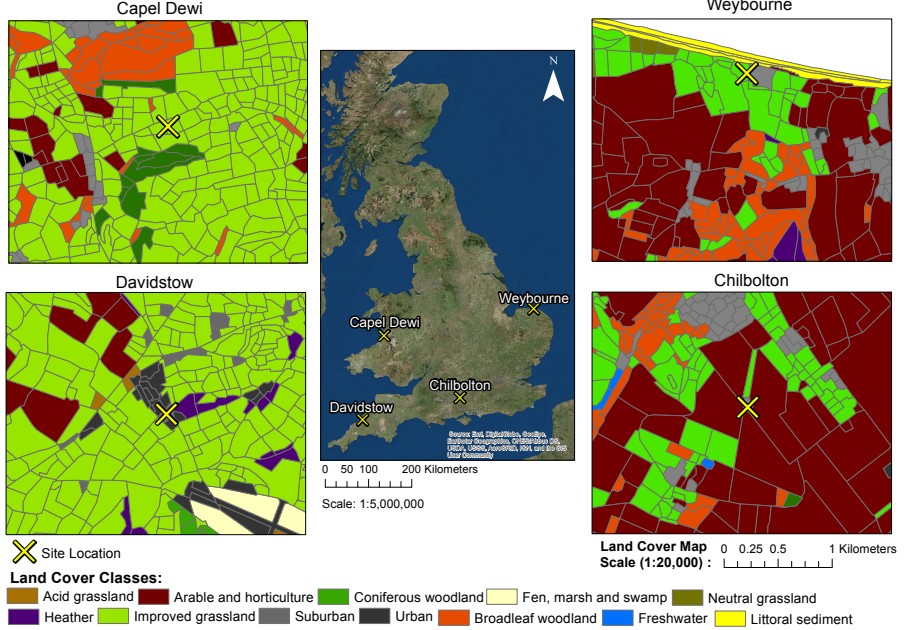

**Figure 1.** Land cover maps per site produced using the Centre for Ecology and Hydrology Land Cover Map 2015 and the World Imagery ArcMap Layer (ArcGIS Version 10.4.1).

grassland land covered area, with some broadleaf woodland to the north of the site and coniferous woodland predominantly to the south, and some arable and horticultural land close by.

Unlike the other sites, Weybourne is a coastal site, located north of Norfolk, at the Weybourne Atmospheric Observatory. Data collection was conducted for a one-week period, from the 17 to the 25 August 2009. The instrument was connected to a high volume sampling line, with an inlet at a height of 10m. The site is located in an improved grassland covered area, with gorse heath observed immediately inland from the Observatory. To the south-east of the Observatory there is arable and horticultural land, to the north of the site there is some littoral sediment, and to the south there is some broadleaf woodland.

## 2.2 UV-LIF Instrumentation

Measurements were recorded at each site using a Wideband Integrated Bioaerosol Spectrometer (Foot et al., 2008; Kaye et al., 2005), with a model 3 (WIBS-3) deployed at Chilbolton Observatory and Weybourne Atmospheric Observatory, and a model 4 (WIBS-4) deployed at Davidstow and the Capel Dewi MST Site. The differences between the WIBS-4 and the WIBS-3 have been previously described and feature different optical chamber design and differing arrangement of the detector wavelength bands (Kaye et al., 2005; Gabey et al., 2010; Crawford et al., 2014; Robinson et al., 2013). A brief summary of the instruments are introduced here, more detailed descriptions have been discussed previously (Gabey et al., 2010; Perring et al., 2014).





The WIBS is able to detect particles ranging from 0.5 to 20μm, encompassing most airborne bacteria and fungal spores, but only very small pollen, or fragments of such. Aerosol particles are drawn from the atmosphere in a laminar flow, and for both the WIBS-3 and WIBS-4 the scattering intensity of a singular particle is measured by a diode laser beam at 635nm in a forward and sideways direction. This is converted to optical equivalent diameter, $D_o$, using a Mie scattering lookup table

which is based on the response of the instrument to calibration PSL spheres. The scattered intensity is measured by a quadrant detector, with the signal from each quadrant used to calculate an average optical diameter over the four scattering angles. Using the four signal intensities, the asymmetry factor ($A_f$) can be determined by calculating the standard deviation between each signal to identify particle morphology. $A_f$ ranges indicate the shape of the particle, as based on measurements using calibration particles. Theoretically an $A_f$ value of 0 indicates a spherical particle, whilst an asymmetrical rod or fibre-like particle yields

an $A_f$ value closer to 100.

Once a particle has been sized, two Xenon flash-lamps are triggered at excitation wavelengths 280nm and 370nm to excite Tryptophan and NADH fluorescence, respectively. Two photomultiplier tubes (PMTs) then record the fluorescence emitted from the particle. Emitted fluorescence is measured using three detector channels which record the fluorescence over two wavelength ranges: FL1 (fluorescence between 300nm and 400nm, once excited at 280nm), FL2 (fluorescence between 410nm

and 650nm, once excited at 280nm), and FL3 (fluorescence between 410nm and 650nm, once excited at 370nm).

## 2.3   Data Sources and Analysis

HAC was used to distinguish and statistically segregate different biological particle types, using the approach of Crawford et al. (2015). Previous studies have reported the success of the method for data analysis in a range of environments (Gabey et al., 2013; Crawford et al., 2014, 2015; Gosselin et al., 2016; Crawford et al., 2017), and the use of such has been compared

to other analysis methods, including various supervised learning techniques (Ruske et al., 2016).

Data pre-processing, prior to clustering the data, is required to remove particles which saturate the PMT, as their true fluorescence cannot be measured. This is also required to remove particles smaller than 0.8μm, as the particle collection efficiency of the WIBS drops below 50% at ~0.8μm. In this study particles are considered fluorescent using a baseline fluorescence (forced trigger measurement) plus 9 standard deviations (SD), contrary to prior studies where 3SD is commonly used (Crawford et al.,

2015; Gosselin et al., 2016; Hernandez et al., 2016). The use of 9SD has been shown in Savage et al. (2017) and used successfully in a comprehensive laboratory study characterising different aerosol materials using a UV-LIF spectrometer. To assess the use of this method for an ambient data set, a comparison between 3SD and 9SD baseline fluorescence is shown for one site to assess the robustness of the HAC approach (Section 3.6).

Each site were analysed as a whole data set including fluorescent and non-fluorescent particles, with the exception of the

Capel Dewi. Here, due to the size of the dataset, fluorescent particles only were selected due to computer memory limitations. The size of each dataset differed, owing to different field campaign durations (Table 1).

The data were normalised using the z-score method, in which the mean is subtracted, and the data is divided by the standard deviation, and the Ward linkage, in which clusters are merged by finding the clusters which yield the minimum increase in total within-cluster variance once merged (Crawford et al., 2015). HAC initially assumes each data point to be its own individual



**Table 1.** The location, and sampling period of each campaign, examples of land cover per site, with the size of the data, number of particle clustered per site, and the instrument used for the campaign presented.

| Location | Sampling period | Land Cover | Dataset size | Particles Clustered | Instrument |
|---|---|---|---|---|---|
| Chilbolton | 20/01/09 - 20/03/09 | Arable, Imp. grassland, woodland | 1.3GB | 406481 | WIBS-3 |
| Davidstow | 25/06/13 - 28/08/13 | Imp. grassland, urban, heather | 5.97GB | 1737369 | WIBS-4 |
| Capel Dewi | 18/02/13 - 03/06/13 | Imp. grassland, woodland | 10.1GB | 3330532 | WIBS-4 |
| Weybourne | 17/08/09 - 25/08/09 | Imp. grassland, arable, littoral sediment | 198MB | 105845 | WIBS-3 |

cluster, and the clusters separated by the shortest distance are combined, until all data points constitute one cluster. The number of clusters to represent the data can be evaluated using the Calinski-Harabasz criterion, which assesses whether dissimilar clusters have been incorrectly combined to form a cluster, and provides the optimum cluster solution to prevent clusters which have small between cluster-variance and large within-cluster variance (Calinski and Harabasz, 1974).

5    Meteorological data were recorded during each campaign, apart from Capel Dewi in which data were obtained from the NERC MST Radar Site. Land cover data, as obtained from the Centre for Ecology and Hydrology Land Cover Map 2015 (LCM2015), were analysed for each site using ArcGIS (Version 10.4.1). Additionally, using the R package, Openair, polar plots of wind direction and wind speed were produced per site to be used in relation to the LCM2015 (Carslaw and Beevers, 2013).

## 3    Results

The average diurnal variation of total fluorescent and non-fluorescent particles were plotted per site (Section 3.1), and following HAC (Section 3.2), the diurnal variation of each cluster was plotted (Section 3.3). The data were then considered in terms of meteorological influences (Section 3.4), focusing on temperature and relative humidity (Section 3.4.1), and wind speed and wind direction (Section 3.4.2).

### 3.1    Total fluorescent particle diurnal variation

The average diurnal variation of fluorescent and non-fluorescent particle concentrations differs between the four sites, potentially indicating the differing types of biological particles present (Fig. 2). Weybourne exhibits increased fluorescent concentrations prior to ~08:00, which increase after ~19:00. At Chilbolton the fluorescent particle concentration remains relatively stable until ~11:00 in which there is a decrease in concentration until ~16:00-17:00 where concentrations slowly increase. Fluorescent concentrations are highest at Davidstow, compared to the other sites, though the trend in fluorescent concentrations throughout the day are less clear, displaying a relatively static trend with the exception of ~13:00 - 20:00 in which there is a marginal drop in fluorescent particle concentration.The fluorescent particle concentrations at Capel Dewi are similar to those at Chilbolton, albeit with slightly less clear diurnal variation. It does appear that there is some small variation during the day, the most apparent trend is the increase in fluorescent particle concentration starting at ~18:00 until 24:00.



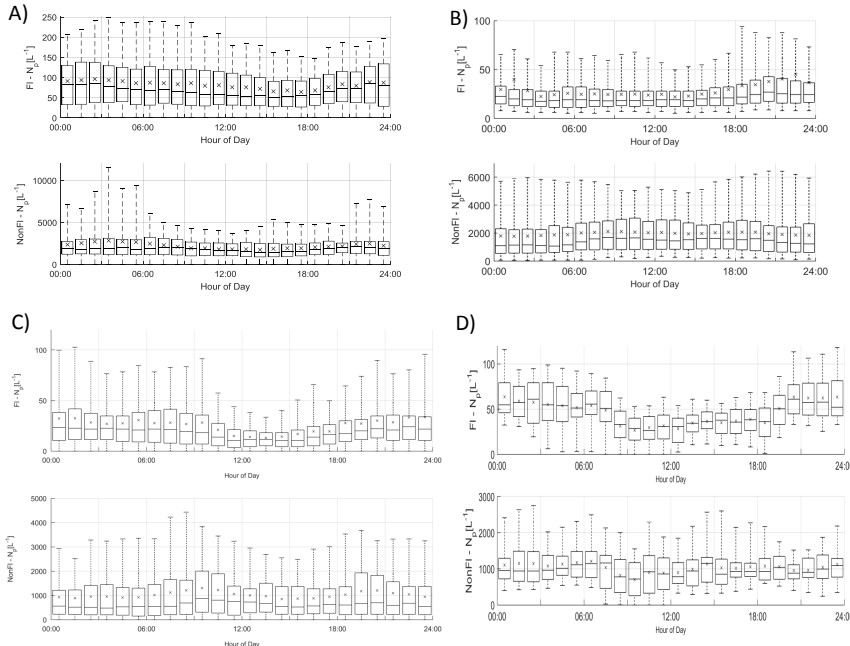

**Figure 2.** Total fluorescent and non-fluorescent particle concentration variation per site [Np L-1] (A - Davidstow; B - Capel Dewi; C - Chilbolton; D - Weybourne)

## 3.2 Cluster Analysis

The standard threshold for defining fluorescent and non-fluorescent particles is calculated by using the instrument forced trigger (FT) measurement ± 3 standard deviations (SD) (Crawford et al., 2015; Gosselin et al., 2016; Hernandez et al., 2016), or 2.5SD (Gabey et al., 2010). Here, the data were analysed using 9SD, which has been found to reduce interference from any other non-biological aerosols and mineral dusts, without compromising the relative fraction of biological particles that are considered fluorescent, as found in a recent comprehensive laboratory study characterising different aerosol materials using a WIBS-4A (Savage et al., 2017). The results of HAC produced a four-cluster solution for both Weybourne and Chilbolton, and a five-cluster solution for Capel Dewi and Davidstow using the Ward linkage and z-score normalisation (Table 2). Using the fluorescent channel intensities, the cluster size and shape, and the percentage contribution to the overall fluorescent particles at each site, initial assumptions were made of the particle type. These assumptions were aided by laboratory experiments conducted utilising a WIBS-3D (Crawford et al., 2017; Ruske et al., 2016), and are hereon referred to as 'Dstl Experiment 2014'. Further support is provided by a comprehensive laboratory experiment conducted at Dstl in 2017 using a WIBS-4, and is hereon referred to as 'Dstl Experiment 2017' (details of which are to be published this year).



**Table 2.** HAC ward linkage analysis per site displaying the average fluorescent intensities per channel FL1, FL2, and FL3; the average optical diameter, D (μm); the average asymmetry factor ($A_f$), and the percentage contribution to the total fluorescent particle count, and inferred cluster type following HAC analysis.

| Site | Cluster | FL1 | FL2 | FL3 | D | $A_f$ | % of Total | Cluster type |
|---|---|---|---|---|---|---|---|---|
| Chilbolton | 1 | 56.0 ± 140.7 | 17.0 ± 50.6 | 213.0 ± 225.5 | 3.6 ± 1.8 | 29.1 ± 12.1 | 23.2 | Fungal |
| | 2 | 11.8 ± 54.7 | 18.0 ± 51.0 | 198.0 ± 207.6 | 1.8 ± 0.7 | 14.4 ± 5.7 | 58.6 | Fungal |
| | 3 | 1005.0 ± 315.0 | 42.3 ± 93.1 | 150.7 ± 203.8 | 3.1 ± 1.3 | 24.9 ± 15.1 | 5.6 | Bacteria/Fungal |
| | 4 | 58.5 ± 203.2 | 474.1 ± 264.7 | 1003.1 ± 360.1 | 2.4 ± 1.6 | 17.9 ± 9.2 | 12.6 | Pollen Fragment |
| Weybourne | 1 | 3.8 ± 25.2 | 12.5 ± 35.6 | 303.6 ± 295.0 | 5.0 ± 2.2 | 36.6 ± 15.9 | 33.7 | Fungal |
| | 2 | 685.5 ± 408.3 | 10.8 ± 37.9 | 120.4 ± 221.4 | 3.1 ± 1.6 | 32.3 ± 16.8 | 6.9 | Bacteria/Fungal |
| | 3 | 5.1 ± 27.3 | 3.2 ± 15.8 | 192.6 ± 200.8 | 2.0 ± 1.1 | 17.2 ± 8.7 | 50.6 | Fungal |
| | 4 | 47.1 ± 190.4 | 292.9 ± 180.3 | 1114.8 ± 345.4 | 4.4 ± 3.2 | 24.5 ± 14.8 | 8.9 | Pollen Fragment |
| Davidstow | 1 | 949.6 ± 711.9 | 1579.2 ± 494.8 | 1671.2 ± 352.5 | 16.5 ± 7.3 | 27.9 ± 16.6 | 0.1 | Pollen Fragment |
| | 2 | 186.9 ± 354.7 | 345.7 ± 278.9 | 271.8 ± 212.5 | 6.5 ± 5.1 | 18.7 ± 16.0 | 4.3 | Interferent |
| | 3 | 87.6 ± 165.7 | 32.5 ± 41.0 | 17.5 ± 32.5 | 5.1 ± 3.1 | 25.8 ± 15.0 | 26.2 | Fungal |
| | 4 | 1534.2 ± 528.3 | 55.3 ± 77.3 | 14.5 ± 38.6 | 4.0 ± 2.0 | 23.0 ± 14.6 | 24.5 | Fungal/Bacteria |
| | 5 | 42.2 ± 115.6 | 22.7 ± 34.1 | 8.0 ± 21.3 | 1.6 ± 0.6 | 6.5 ± 2.7 | 44.9 | Fungal |
| Capel Dewi | 1 | 455.4 ± 422.9 | 1197.0 ± 435.6 | 834.8 ± 308.9 | 5.3 ± 4.9 | 15.8 ± 13.6 | 0.8 | Interferent |
| | 2 | 123.2 ± 198.8 | 385.9 ± 245.1 | 171.6 ± 126.2 | 1.8 ± 0.8 | 7.0 ± 3.8 | 8.7 | Interferent |
| | 3 | 1070.3 ± 471.5 | 30.8 ± 71.6 | 10.8 ± 38.3 | 3.4 ± 2.1 | 21.7 ± 16.9 | 10.3 | Fungal/Bacteria |
| | 4 | 78.7 ± 143.9 | 48.5 ± 94.5 | 36.6 ± 93.8 | 6.1 ± 3.3 | 28.0 ± 13.7 | 18.7 | Fungal |
| | 5 | 71.6 ± 124.2 | 34.0 ± 53.1 | 7.3 ± 21.1 | 1.6 ± 0.8 | 7.7 ± 4.3 | 61.5 | Fungal |

### 3.2.1 Weybourne

The dominant cluster at Weybourne is Cluster 3, representing 50.6% of total fluorescent particle concentration, and exhibiting the greatest fluorescence signal in channel FL3, similar to Cluster 1. Cluster 1 displays a similar fluorescence profile and similar high abundance (33.7% of total fluorescent particle concentration), however, with a much larger size and shape (5μm, $A_f$ 36.6) compared to Cluster 3, which exhibits the smallest size and shape compared to other clusters (2μm, $A_f$ 17.2). Though both clusters show similar fluorescent signals, it is likely that due to the differences in size and shape values, these clusters have been segregated. Laboratory data collected using a WIBS-3 have shown that bacteria such as unwashed E-Coli and Bacillus atrophaeus (BG) spores exhibit higher fluorescence in channel FL3 (Dstl Experiment 2014). This is contrary to other studies which have found a strong FL1 signature for bacteria (Crawford et al., 2017; Hernandez et al., 2016; Savage et al., 2017). Given the abundance of these two clusters, it seems unlikely that these clusters are bacteria, as rates of bacteria have been found to be diverse but low at coastal sites (Shaffer and Lighthart, 1997). As these two clusters represent the two largest clusters at this site, it is possible that these may be fungal spores, as fungal spores have been found to be the most prevalent airborne



biological particles compared to other bioaerosol classes (Elbert et al., 2006; Fisher et al., 2012; Fröhlich-Nowoisky et al., 2009) . Although, the fluorescent profiles of these clusters do not match previous literature of fungal spores, in which there is an FL1 dominance (Hernandez et al., 2016; Healy et al., 2012; Gabey et al., 2013; Savage et al., 2017).

Cluster 2 is the least abundant cluster and represents only 6.9% of the total fluorescent particle concentration. Fluorescence signal is greatest in channel FL1, with some signal in channel FL3, and very low signal in channel FL2. The size and shape of this cluster indicate a fairly small, aspherical particle (3.1μm, $A_f$ 32.3). Higher FL1 concentrations and lower median FL2 and FL3 values have been associated with bacteria and fungal spores (Savage et al., 2017), however, in a laboratory study using the WIBS-4, compared to bacterial samples, the fungal spores Alternaria and Cladosporium were found to have a higher FL1 channel in comparison to channels FL2 and FL3 (Dstl Experiment 2017). It is possible that this is a fungal spore, however owing to the small average particle size, and low abundance, it is also possible that this is a bacterial particle or aggregate containing bacterial cells. Previously, bacteria size ranges has been found to be 4μm, with fairly high asymmetry factors, similar to this cluster (Gabey et al., 2011, 2013).

Cluster 4 also accounts for a small proportion of fluorescent particles (8.9% of the total) but displays high fluorescence in channel FL3, with some fluorescence in FL2, and less in FL1. Particle size is relatively large, and similar to the other clusters, the shape value indicates an aspherical particle (4.4μm, $A_f$ 24.5). The high fluorescence signal in channel FL3 is similar to Cluster 1 and Cluster 3, yet for Cluster 4 there is moderate signal in FL2 and some signal in FL1. It is possible that this is a pollen fragment, given the time of year and pollen season, and as a larger FL3 values have been associated laboratory sampled Ryegrass, Aspen, and Poplar pollen (Dstl Experiment 2014). However, often there will be a similar high FL2 value (O'Connor et al., 2014; Crawford et al., 2017), which does not occur here for Cluster 4. Using the fluorescent profiles for each cluster at Weybourne, it is initially suggested that these comprise fungal spores (Cluster 1 and Cluster 3), bacteria (Cluster 2), and potential pollen fragments (Cluster 4).

### 3.2.2 Chilbolton

Cluster analysis of the Chilbolton dataset similarly produced a four-cluster solution, of which, Cluster 2 accounts for 58.6% of the fluorescent particle concentration. Cluster 2 displays low fluorescence in channels FL1 and FL2, with higher fluorescence in channel FL3. Not only is this fluorescence profile and cluster dominance similar to Weybourne Cluster 3, Chilbolton Cluster 2 also exhibits the smallest size and shape (1.8μm, $A_f$ 14.4).

Chilbolton Cluster 1 displays a similar fluorescence profile to Chilbolton Cluster 2. However, Chilbolton Cluster 1 is less abundant (23.2%), and exhibits the largest size and shape (3.6μm, $A_f$ 29.1). Similar to Weybourne, it is likely that these clusters are a subset of the same particle type, segregated due to differences in size and asymmetry factor. Incidentally, the fluorescent profile of these clusters are remarkably similar to Weybourne Cluster 1 and Cluster 3, and though Chilbolton Cluster 1 shows some fluorescence in channel FL1, it is possible that these clusters are fungal spores.

Chilbolton Cluster 3 is the smallest cluster (representing 5.6% of the total fluorescent population) but exhibits the greatest fluorescence in channel FL1 compared to the other clusters, and exhibits a similar fluorescent profile to Weybourne Cluster 2. Both Weybourne Cluster 2 and Chilbolton Cluster 3 make up the smallest cluster group of each dataset, with the same size



(3.1µm) but slightly differing $A_f$ values of 24.9 and 32.3 for Chilbolton, and Weybourne, respectively. It is likely that these two clusters are the same type of particle, and are representative of bacterial or fungal spores, as discussed for Weybourne Cluster 2.

Cluster 4 from the Chilbolton dataset displays similar fluorescent profile characteristics to Weybourne Cluster 4, in which there is a high fluorescence signal in channel FL3, with some in FL2 and less so in FL1. Chilbolton Cluster 4 represents 12.6% of the total fluorescent particle population, and has a size of 2.4µm and shape $A_f$ 17.9. Though Weybourne differs in terms of size and shape (4.4µm, $A_f$ 24.5), the fluorescence profile of these clusters is inferred to be representative of pollen fragments.

The fluorescent profiles at Chilbolton show much similarity to Weybourne, even though Weybourne is a coastal site. This suggests that the sources of these clusters originate from the same land cover type, which for both sites is either the improved grassland or arable and horticultural land. Using the fluorescent profiles for each cluster, the clusters at Chilbolton are initially assumed to comprise fungal spores (Cluster 1 and Cluster 2), bacterial or fungal spores (Cluster 3), and a pollen fragment (Cluster 4).

### 3.2.3 Capel Dewi

Cluster analysis of the MST Capel Dewi dataset produced a five-cluster solution, of which, Cluster 5 accounts for 61.5% of the total fluorescent cluster population. Cluster 5 exhibits relatively low fluorescent channel signals in comparison to other clusters, with the exception of Cluster 4 which presents a similar fluorescent profile, and represents 18.7% of the total fluorescent particle population, the second largest cluster. Incidentally, Cluster 5 represents the smallest size (1.6µm) in comparison to Cluster 4 which represents the largest average size (6.1µm). Similarly, there is also some difference between the asymmetry factor between these two clusters with a more spherical particle for Cluster 5 ($A_f$ 7.7), and a non-spherical particle for Cluster 4 ($A_f$ 28). It seems likely that Cluster 4 is a subset of Cluster 5, segregated due to differences in particle size and asymmetry factor. It can be speculated that this is a fungal spore, given the low fluorescence signal intensity and moderate difference between each fluorescent signal, as well as the high intensity in channel FL1, and overall cluster abundance (Savage et al., 2017; Hernandez et al., 2016). Additionally, recent laboratory experiments have shown low fluorescent signals, but higher fluorescence in channel FL1 for Cladosporium and Alternaria samples (Dstl Experiment 2017).

Cluster 3 exhibits a large fluorescence signal in channel FL1, with only some in FL2, and less so in channel FL3. The larger fluorescence in channel FL1 is similar to Chilbolton Cluster 3 and Weybourne Cluster 2. Though these two sites have similar signals in channels FL2 and FL3, Capel Dewi Cluster 3 signals decrease in channel FL2 and even more so in channel FL3. However, comparing the size range of the particles between these sites, they are remarkably similar, with a size range of 3.1µm (Chilbolton and Weybourne) to 3.4µm (Capel Dewi). The asymmetry factors between these sites are additionally in agreement and indicate an asymmetrical particle (Capel Dewi -$A_f$ 21.7; Chilbolton - $A_f$ 24.9; Weybourne - $A_f$ 32.2). This cluster type is least prominent at Chilbolton and Weybourne, however at Capel Dewi this represents 10.3% of the total fluorescent population. Owing to the large fluorescent signal in channel FL1, it is speculated that this Cluster is a fungal spore or bacterial particle.

Cluster 2 represents 8.7% of the total fluorescent population, and displays a high fluorescence signal in channel FL2, with some signal in channel FL1 and FL2. This cluster shows some similarities to Cluster 1, the smallest cluster (0.8% of the total



fluorescent cluster population). Though Cluster 1 displays a higher fluorescent signal than Cluster 2, a similar fluorescence signal profile is noticeable. The sizes of these two clusters differ, with Cluster 2 reasonably small (1.8μm) and spherical ($A_f$ 7), and Cluster 1 larger (5.3μm) and aspherical ($A_f$ 15.8). A larger fluorescence signal in channel FL2 (or type B) has been associated with interferent particles (Hernandez et al., 2016), though the fluorescence signal in channel FL1 and FL3 does not imply that these particles are interferent.

The cluster types at Capel Dewi are suggested to comprise fungal spores (Cluster 4 and Cluster 5), and it is inferred that Clusters 1 and 2 are interferents given the larger signal in channel FL2. Cluster 3 is assumed to be a fungal spore or bacterial particle, which shows some slight similarity to Chilbolton Cluster 3 and Weybourne Cluster 2 in terms of the high FL1 channel.

### 3.2.4 Davidstow

Similar to the Capel Dewi dataset, cluster analysis of the Davidstow data produced a 5-cluster solution. Of which, Cluster 5 represents 44.9% of the total fluorescent particle population, but displays a very low fluorescent signal and small size and shape (1.6μm, $A_f$ 6.5), indicating a small, spherical particle. The fluorescent signal of Cluster 5 is similar to Cluster 3 which represents the second largest cluster (26.2% of the total fluorescent population), though with a larger size and shape (5.1μm, $A_f$ 25.8) than Cluster 5. These clusters are similar to two of the most prominent clusters at Capel Dewi, Clusters 4 and 5, with similar sizes and asymmetry factors between both sites for Cluster 5 (Capel Dewi - 1.6μm, $A_f$ 7.7; Davidstow - 1.6μm, $A_f$ 6.5), and near similar for Capel Dewi Clusters 4 (6.1μm, $A_f$ 28) and Davidstow Cluster 3 (5.1μm, $A_f$ 25.8). The very low fluorescent signals, with slightly higher signal in channel FL1 indicates that like Capel Dewi Clusters 4 and 5, these two clusters are fungal spores.

Again, similarities between Davidstow Cluster 4 and Capel Dewi Cluster 3 can be seen, though, additionally with some similarities to Weybourne Cluster 2 and Chilbolton Cluster 3, in terms of the high fluorescence signal in channel FL1. Davidstow Cluster 4 and Capel Dewi Cluster 3 display a very similar fluorescence signal profile, albeit at different fluorescent intensities. Davidstow Cluster 4 represents 24.5% of the total fluorescent particle population, with an average size and shape (4μm and $A_f$ 23) which is similar to Capel Dewi Cluster 3 (3.4μm and $A_f$ 21.7), it is therefore suspected that this is also a fungal spore or bacterial particle.

Davidstow Cluster 2 displays a similar fluorescent profile to Capel Dewi Cluster 2, and represents a small proportion of the fluorescent particle population (4.3%) similar to Capel Dewi Cluster 2 (8.7%). However, there are differences in size and shape, as Davidstow Cluster 2 has a larger average particle size and asymmetry factor compared to Capel Dewi Cluster 2. It can be seen that Davidstow Cluster 2 also displays some similarity to Capel Dewi Cluster 1, though at lower fluorescence signal intensities, but similar in terms of average particle size and asymmetry factor (Capel Dewi Cluster 1 - 5.3μm and $A_f$ 15.8). Considering this fluorescence profile, it is possible that the higher fluorescence in channel FL2, as discussed previously for Capel Dewi Clusters 1 and 2, indicates that this cluster comprises interferent particles.

Davidstow Cluster 1 is a remarkable cluster, owing to its large size (16.5μm), and low abundance (0.1%). This cluster exhibits a consistent shape in relation to other sample sites ($A_f$ 27.9), but has high fluorescent signals in each channel, with the highest in Channel FL3. This cluster is similar to Chilbolton Cluster 4 and Weybourne Cluster 4, in terms of the large signal





**Table 3.** Summary table of similar Clusters (CL) between sites with the inferred cluster grouping and type

| Group 1 Fungal | Group 2 Fungal | Group 3 Bacteria/Fungal | Group 4 Fungal/Bacteria | Group 5 Pollen Fragment | Group 6 Interferent |
|---|---|---|---|---|---|
| Davidstow - CL3 | Chilbolton - CL1 | Chilbolton - CL3 | Davidstow - CL4 | Chilbolton - CL4 | Davidstow - CL2 |
| Davidstow - CL5 | Chilbolton - CL2 | Weybourne - CL2 | Capel Dewi - CL3 | Davidstow - CL1 | Capel Dewi - CL1 |
| Capel Dewi - CL4 | Weybourne - CL1 | | | Weybourne - CL4 | Capel Dewi - CL2 |
| Capel Dewi - CL5 | Weybourne - CL3 | | | | |

in FL3, with some in FL2, and lesser in channel FL1. However, the fluorescence intensity and size makes this cluster unique. Considering the fluorescent signal and low abundance of this cluster, this may well be a pollen fragment, especially considering the larger average size of this cluster. The clusters at Davidstow show much similarity to the Capel Dewi dataset. The inferred cluster types present include fungal spores (Cluster 3 and Cluster 5), pollen fragments (Cluster 1), interferents (Cluster 2), and

either a fungal spore or bacterial particle (Cluster 4).

### 3.2.5   Grouping of Clusters

Though HAC proved successful in segregating the fluorescent particle data into distinctive clusters, the method alone does not allow for easy particle identification, as demonstrated during the first attempt to assign particle type to each cluster. For ease of analysis, a summary table of the similar clusters is presented (Table 3), identifying the clusters which share a similar

fluorescent profile between the four different sites, and were speculated to belong to the same particle type.

### 3.3   Cluster Diurnal Variation

The average diurnal variation of each cluster per site were plotted to identify any diurnal patterns which influence cluster concentration at each hour of day (Fig. 3). These have been collated together according to the suggested groupings made based on HAC analysis (Table 3), rather than per site, in order to assess the ability of characterising particles based on fluorescent

signal.

*Group 1 - Fungal Spore*

Following HAC, the clusters comprising Group 1 were considered fungal spores. Both Davidstow Cluster 3 and Cluster 5 show a fairly static diurnal concentration, indicative that these are the same particle class. The stability in diurnal concentrations is also experienced for Capel Dewi Cluster 4 and Cluster 5, in which both clusters show small diurnal variation, and remain

relatively static. One exception to this is that Cluster 5 concentrations increase after 4pm, whereas Cluster 4 very gradually increases in concentration throughout the day, and then decreases after 9pm. It can be concluded that these clusters have been accurately collected together and can be assumed to be the same type of particle. However, the behaviour of these clusters do not reflect typical fungal spore behaviour, and instead show some characteristics of bacterial particles given the lack of diurnal variation.





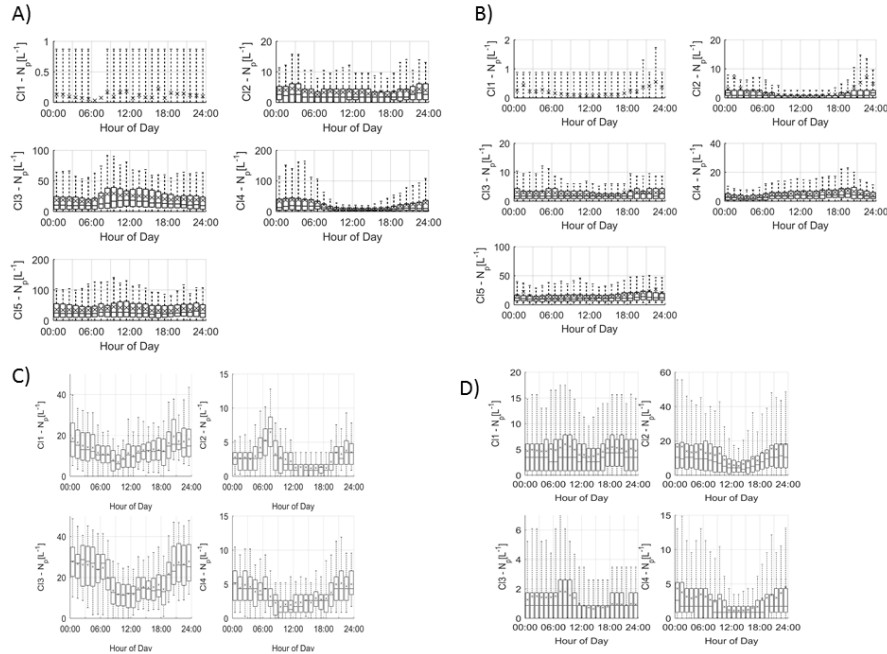

**Figure 3.** Average cluster diurnal variation per site (A - Davidstow, B - Capel Dewi, C - Weybourne, and D - Chilbolton)

*Group 2 - Fungal Spore*

The clusters comprising Group 2 were suspected to be fungal spores. Both Chilbolton Cluster 1 and Cluster 2 exhibit a mid-day decrease in concentrations from ~9am - 10am until 3pm - 4pm, with Cluster 2 displaying a pronounced mid-day decrease compared to Cluster 1. This is similar to Weybourne Cluster 3 which exhibits a decline in concentration from 12pm - 9am, and

5    remains low during the mid-morning, but very generally increases from 1pm - 12pm. Weybourne Cluster 1 does not experience a mid-day decline in concentrations and instead a decrease in concentration occurs from 12pm - 9am, and increases gradually throughout the day, until 9pm in which concentrations become more stable. With the exception of Weybourne Cluster 1, the clusters in this group share similar diurnal patterns, which appears to be representative of fungal spore behaviour.

*Group 3 - Bacteria or Fungal Spore*

10   The two clusters in Group 3 experience an increase in concentrations before 9am, which is more pronounced and gradual for Weybourne Cluster 2. This is followed by a decline in concentrations from around 12pm - 1pm until 6pm for both Chilbolton Cluster 3 and Weybourne Cluster 2. An increase in concentrations from 6pm onwards can be observed, with Weybourne exhibiting a gradual increase each hour until midnight. The decline in concentrations during the day, especially during warmer temperatures which occur after 12pm, is indicative of fungal spore activity. However, the cause of the increase in concentrations

15   before 9am, especially for Weybourne, is unclear.

*Group 4 - Fungal Spore or Bacteria*  Owing to the high fluorescence signal in channel FL1 Group 4 was assumed to be either a fungal spore or bacterial particle. Davidstow Cluster 4 shows the most distinct pattern of any cluster, with a gradual decrease





in concentration until 8am, followed by low concentrations from 9am - 6pm, from which concentrations gradually increase. However, the trend for Capel Dewi Cluster 3 is not as distinct, with a very small diurnal variation. The diurnal profiles of these two clusters do not match, even though the fluorescent profiles are similar. However, It is suggested that these clusters are both fungal spores, and it is proposed that due to the size of the Capel Dewi dataset that the diurnal variation for this cluster is not

as clear.

*Group 5 - Pollen Fragment*

This group of clusters were considered to be pollen fragments, owing to the low contribution to the total fluorescent particle population, and high fluorescence in channel FL3, which was accompanied by high (Davidstow Cluster 1) to moderate (Chilbolton Cluster 4 and Weybourne Cluster 4) fluorescence in channel FL2. The trend is similar for both Chilbolton and

Weybourne Cluster 4, in which there is a drop in concentrations during the morning, with lower concentrations during the day, which then increase in the evening. This illustrates that these are fungal spores, unlike Davidstow Cluster 1, which shows more variation during the day. Considering this, it is possible that only Davidstow Cluster 1 is a pollen fragment, and Chilbolton and Davidstow Cluster 4 are both fungal spores. This potentially illustrates that the dominance of channel FL2, when accompanied by a high signal in channel FL3, is what separates pollens from fungal spores.

*Group 6 - Interferent*

Previous literature has described a dominant signal in channel FL2 to be representative of interferent particles, however, for Group 6 the dominance in channel FL2 is accompanied by moderate-to-high signal in channel FL3, and slightly lower signal in channel FL1. The diurnal variation of Capel Dewi Cluster 1 and Cluster 2 is reminiscent of other clusters which have been suggested to be fungal spores, owing to the lower concentrations throughout the day, and the decrease in concentrations

in the morning, and subsequent increase in concentrations in the early evening. The identity of Davidstow Cluster 2 though is uncertain, owing to the lack of fluctuations throughout the day, and a fairly static concentration profile, therefore further analysis with the meteorological data is required.

## 3.4   Meteorological Influences

The influence of local scale meteorology, in particular the relationships between cluster abundance and temperature (°C), and

relative humidity (RH %) were investigated in relation to bioaerosol abundance (as discussed in Section 1), but focusing on the cluster groupings between sites (Table 3), and not on a per site basis.

### 3.4.1   Temperature and Relative Humidity

*Total fluorescent particle concentrations*

The relationship between total fluorescent particles and temperature and relative humidity (RH) at Weybourne show an

increase in concentration with increased RH, and a decrease with increased temperature. Similarly, total fluorescent particle concentrations increase with increased RH at Chilbolton, and show a decrease in concentrations at temperatures >5°C. The behaviour for these sites is indicative of wet discharged fungal spores, and reflects the average diurnal activity of fluorescent particles at each site. Davidstow differs from these sites and instead experiences a decrease in concentration when RH >60% but





an gradual increase in concentration with increasing temperature, decreasing slightly at temperatures >25°C, which is indicative of bacterial particles. In contrary, Capel Dewi does not present a clear trend with temperature and relative humidity. A slight increase in concentration does appear once RH >80% and concentrations appear to gradually increase with temperature, with lower concentrations experienced at 10 - 15°C, and a decrease in concentration occurring at >20°C.

*Group 1 - Fungal Spore/Bacteria (Davidstow CL3 and CL5; Capel Dewi CL4 and CL5)*

The decrease in concentration for Davidstow Cluster 3 and Cluster 5 when RH is >60% and temperatures are >25°C indicates this may be a bacterial particle as optimum temperatures for bacterial production have been found to be ~22°C in a salt marsh estuary (Apple et al., 2006). However, the decrease in concentrations above this temperature may indicate that this is a fungal spore, in particular, Cladosporium, which has been found to be most abundant at temperatures of 20°C to 24°C

(Fernández-Rodríguez et al., 2017). Though temperatures at Capel Dewi were lower than Davidstow, and the trend is less clear, it can be seen that Capel Dewi Cluster 4 shows a similar trend to Davidstow Cluster 3 and 5, and it is suggested too that this is either the fungal spore, Cladosporium, or a bacteria particle. Analysing the diurnal cluster timeseries, concentrations are higher during the day, and bacterial particle concentrations have been previously found to be lower between 9pm - 5am (Shaffer and Lighthart, 1997), as a result of trapping of bacteria by the inversion layer and gradual accumulation throughout the

day (Abdel Hameed et al., 2009). However, Capel Dewi Cluster 5 does not show a clear trend, and differs from other clusters in this group. It is possible that this is a wet discharged fungal spore, owing to the increased concentration when RH increases, and decrease in concentration when temperature increases.

     *Group 2 - Fungal Spores (Chilbolton CL1 and CL2; Weybourne CL1 and CL3)*

The temperatures at Chilbolton were colder than those at Weybourne, and the prevalence of these clusters when tempera-

tures are low (with the exception of Weybourne Cluster 1) and the general increase of these clusters with RH suggests that Chilbolton Cluster 1 and Cluster 2, and Weybourne Cluster 3, are wet discharged fungal spores. This is emphasised by the cluster diurnal timeseries, which, for the most part, displays higher concentrations in the early morning and evening. In comparison, Weybourne Cluster 1 does not exhibit a similar diurnal pattern, and does not show any particular trend with increased temperature, nor do concentrations immediately increase with increased RH, until RH is >70%. It cannot be assumed that this

is a wet discharged fungal spore, and instead the cluster type is considered, at present, to be unknown.

     *Group 3 - Fungal Spores (Chilbolton CL3 and Weybourne CL2)*

Group 3 is inferred to comprise wet discharged fungal spores, owing to the decrease in concentrations when temperatures are >10°C and >15°C, for Chilbolton Cluster 3 and Weybourne Cluster 2, respectively. Though Weybourne Cluster 2 exhibits a decrease in concentrations at RH >80%, the general increase in concentrations with increasing RH further suggests that these

are wet discharged fungal spores.

     *Group 4 - Fungal Spore (Davidstow CL4 and Capel Dewi CL3)*

Temperatures at Capel Dewi were colder than those at Davidstow, and concentrations increase until temperatures reach 5 - 10°C. At temperatures >10°C the trend is less clear, but do decrease slightly, with the exception of a peak occurring at 10 - 15°C. Similarly, concentrations for Davidstow Cluster 4 peak at 15°C, and at temperatures above these, the concentration

decreases. Both clusters show similar RH patterns, with stable concentrations until an RH of >70% and >80% for Davidstow




and Capel Dewi, respectively. These characteristics are somewhat similar to those experienced for Group 2 and Group 3, and it is inferred that this group are wet discharged fungal spores.

*Group 5 - Pollen Fragment (Davidstow CL1) / Fungal Spore (Chilbolton CL4; Weybourne CL4)*

This group were assumed to consist of pollen fragments, however, following analysis of the cluster diurnal profiles, Chilbolton Cluster 4 and Weybourne Cluster 4 were assumed to be fungal spores. It can be inferred that these two clusters are wet discharged fungal spores as there is a decrease in concentration at temperatures >5°C for Chilbolton, and >15°C for Weybourne. Similarly, there is an increase in concentration with increased RH, with Weybourne Cluster 4 doing so once RH is >50-60%. Davidstow Cluster 1 does not display a similar trend, instead concentrations increase with increased temperature until >25°C, and decrease with increasing RH. Considering this, it is deduced that this behaviour is representative of bacteria, especially given the diurnal cluster pattern in which some higher concentrations are experienced during the day.

*Group 6 - Interferent (Davidstow Cluster 2; Capel Dewi Cluster 1 and Cluster 2)*

The cluster type of Davidstow Cluster 2 was unknown following analysis of the average cluster diurnal variation. However, the concentrations of this cluster increase at temperatures >15°C, and initially increase until RH >60%, in which a decrease in concentration occurs, which is suggestive of bacteria behaviour (Fig. 4). Capel Dewi Cluster 1 and Cluster 2 are dissimilar to Davidstow Cluster 2, and though most average values are outside of the boxplot range, an increase in concentration occurs with increasing RH, especially for Capel Dewi Cluster 2 (Fig. 4). Similarly, concentrations increase until temperatures are >5°C, after which concentrations steadily decrease for Cluster 2. Similar to other groups, this behaviour is indicative of wet discharged fungal spores, such as Ascospores or Basidiospores.

### 3.4.2 Wind Speed and Wind Direction

In the previous section, the data were split according to the fluorescent signal profiles, using HAC analysis to initially assign a cluster class, and cluster diurnal timeseries and meteorological data to confirm or contradict these assumptions (Table 4). Following this, the relationship between total fluorescent particles and clusters at each site were analysed as a function of wind speed (ms⁻¹) and wind direction using the LCM2015 per site to identify the potential sources, and the abundance of different particulates between different land covered areas.

Weybourne

The sources of the total fluorescent particles at Weybourne appear to originate from a south-southwest (SSW) direction, at low wind speeds (2 - 4 m s⁻¹) (Fig. 5) . Each cluster has higher concentrations from the SSW wind direction, at wind speeds of a fairly similar intensity as for total fluorescent particles (~2 - 4 m s⁻¹). Cluster 1 additionally shows a source originating from a south-westerly (SW) direction at higher wind speeds of 10 - 12 m s⁻¹ (Fig. 6). Considering the presence of fairly local fluorescent particles, the sources for all the clusters are likely to originate from the 'Improved grassland' covered areas, which is likely to lead to higher fungal spore concentrations, and it can be inferred that there is an association between grassland areas and wet discharged fungal spores. The exception to this is Cluster 1, which may comprise fungal spores, but also particles from arable and horticultural land SW of the site. It is thereby possible that this cluster is a mixture of two different particle types,



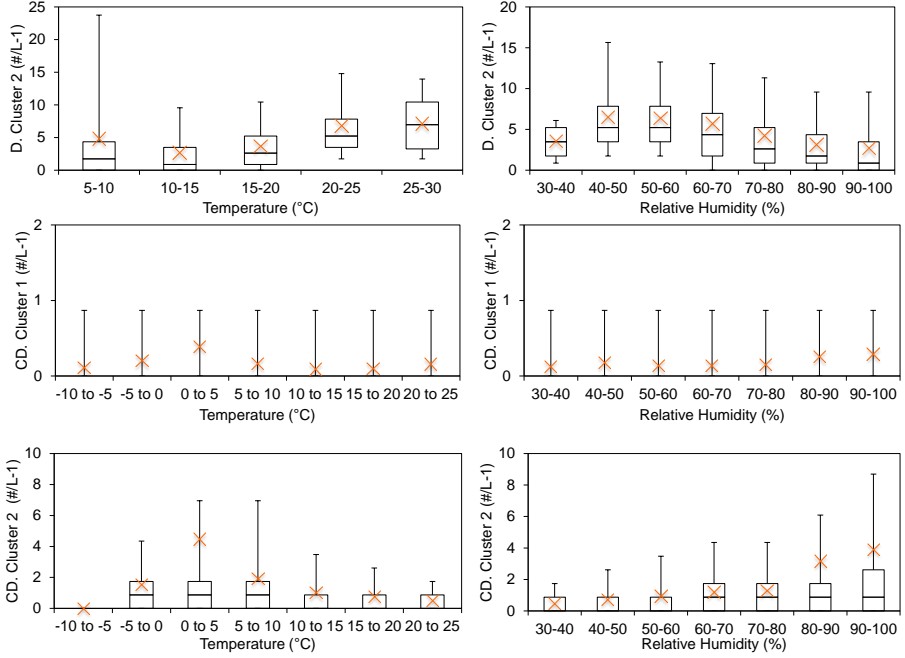

**Figure 4.** Group 6 were considered to be non-biological interferent particles following HAC analysis, following Hernandez et al. (2016) in which FL2 (or type B) was the dominant signal. Instead, Davidstow Cluster 2 was deemed to be a bacterial particle, and Capel Dewi Cluster 1 and Cluster 2 wet discharged fungal spores following temperature (left) and relative humidity (right) analysis. (Site names shortened for brevity D - Davidstow, CD - Capel Dewi.)

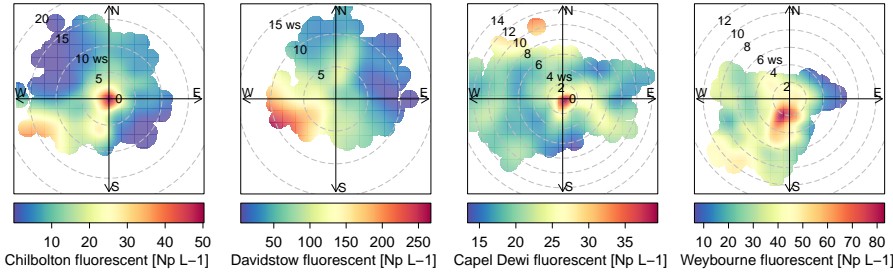

**Figure 5.** Total fluorescent particle concentration per site for each campaign period [Np L$^{-1}$]

with one originating from the arable and horticultural land, and the other comprising fungal spores, which complicates particle identification.

Chilbolton




**Table 4.** Suspected particle class types per site showing the inferred cluster types following fluorescence signal analysis, and the suggested particle types following the cluster diurnal time series, and meteorological data analysis (temperature and relative humidity). (Note: Weybourne and Chilbolton comprise 4 clusters, hence unassigned Cluster 5)

| FL Signature | CL1 | CL2 | CL3 | CL4 | CL5 |
|---|---|---|---|---|---|
| Weybourne | Fungal Spore | Bacteria/Fungal | Fungal Spore | Pollen Fragment | - |
| Chilbolton | Fungal Spore | Fungal Spore | Bacteria/Fungal | Pollen Fragment | - |
| Capel Dewi | Interferent | Interferent | Fungal/Bacterial | Fungal Spore | Fungal Spore |
| Davidstow | Pollen Fragment | Interferent | Fungal Spore | Fungal/Bacterial | Fungal Spore |
| **Diurnal Variation** | **CL1** | **CL2** | **CL3** | **CL4** | **CL5** |
| Weybourne | Unclassified | Fungal | Fungal | Fungal | - |
| Chilbolton | Fungal | Fungal | Fungal | Fungal | - |
| Capel Dewi | Fungal | Fungal | Fungal | Bacteria | Bacteria |
| Davidstow | Pollen Fragment | Unclassified | Bacteria | Fungal | Bacteria |
| **Met. Analysis** | **CL1** | **CL2** | **CL3** | **CL4** | **CL5** |
| Weybourne | Mixture | WD Fungal | WD Fungal | WD Fungal | - |
| Chilbolton | WD Fungal | WD Fungal | WD Fungal | WD Fungal | - |
| Capel Dewi | WD Fungal | WD Fungal | WD Fungal | Bacteria | WD Fungal |
| Davidstow | Bacteria | Bacteria | Bacteria | WD Fungal | Bacteria |

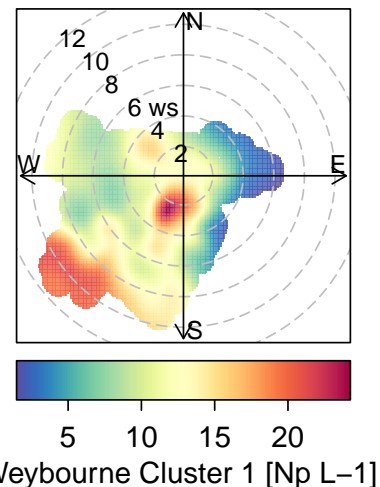

**Figure 6.** Weybourne Cluster 1 polar plot, presenting two potential sources of biological particles, which when using Figure 1, are likely to comprise improved grassland and arable and horticultural land.



Total fluorescent particle concentrations at Chilbolton originate from a local source at low wind speeds, with lower concentrations of particles originating from a west-southwest (WSW) direction at wind speeds of ~12 - 20 m s$^{-1}$ (Fig. 5). Concentrations from the WSW direction can be seen for Cluster 1 at wind speeds >10 m s$^{-1}$, in which, incidentally, there resides a composting and mushroom farm, a probable source of fungal spores. Lower concentrations from the WSW wind direction can be seen for

Cluster 2, however, most particles predominantly originate locally, from a mixture of arable and horticultural land, and some small improved grassland. Cluster 4 comprises particles from a local source, and similarly as does Cluster 3, however from a north and north-westerly direction at wind speeds of <5 m s$^{-1}$. The surrounding land of Chilbolton Observatory comprises arable and horticultural land, with some improved grassland cover. Given the low wind speeds and local sources for Cluster 2, Cluster 3, and Cluster 4, it is likely that emissions from the surrounding land is the source of the suspected wet discharged

fungal spores.

Davidstow

The source of fluorescent particles at Davidstow originate from a WSW wind direction, at wind speeds from ~5 - 10 m s$^{-1}$ (Fig. 5). Cluster 2, Cluster 3, and Cluster 5 show increased concentrations from this WSW direction and wind speed, with Cluster 2 and Cluster 5 exhibiting higher concentrations from a SW wind direction. These clusters originate from predominantly

improved grassland land cover, but with some arable and agricultural land, and some urban and suburban land. These clusters were assumed to comprise bacterial particles, and given the similarity in wind direction and wind speed for these three clusters, it is likely that they originate from the same source. Cluster 4 was assumed to comprise wet discharge fungal spores, and this cluster shows high concentrations from a north/north-northeasterly (NNE) wind direction, at wind speeds of ~4 - 10 m s$^{-1}$, which comprises both improved grassland and suburban areas, with a large cheese factory roughly north-west of the site.

Cluster 1 has high concentrations of particles originating from a south-easterly wind direction at wind speeds of mostly <5 m s$^{-1}$, The land cover comprises urban land, with some heather, and predominant improved grassland. Similar to Clusters 2, 3, and 5, Cluster 1 was suggested to comprise bacterial particles, which may originate from the urban area, as bacterial concentrations have been found to be higher in urban areas compared to other sites (Shaffer and Lighthart, 1997). However, the clusters which were assumed to comprise bacteria do not have a typical expected size range for bacterial particles, it is possible that these

have been combined with another particle.

Capel Dewi

The source of total fluorescent particles at Capel Dewi mostly originate locally, with some contribution from a north-northwesterly (NNW) wind direction at wind speeds of 10 - 14 m s$^{-1}$ (Fig. 5). Cluster 5 comprises the largest cluster at Capel Dewi, and originates locally at a wind speed of <2 m s$^{-1}$, from a southerly wind direction. There is also some very small

contribution from an easterly wind direction at a wind speed of 2 - 14 m s$^{-1}$, and an additional small source from a NNW wind direction at speeds of 8 - 12 m s$^{-1}$. Cluster 5 and Cluster 4 were initially assumed to be wet discharged fungal spores owing to the similar fluorescence profile and abundance, however, following meteorological data analysis of Cluster 4 it was inferred this was a bacterial particle. Cluster 4 is distinct and exhibits no apparent main source of fluorescent particles, though higher concentrations can be seen to originate from a NNW wind direction, at similar wind speeds to Cluster 5. This cluster

may originate from the broadleaf and coniferous woodland to the NNW of the site, as alongside urban sites, forests have also





**Table 5.** R values fitted to third-order polynomial fits between total fluorescent particles and clusters per site, and the meteorological variables, temperature and relative humidity

| Weybourne | FL | CL1 | CL2 | CL3 | CL4 | |
|---|---|---|---|---|---|---|
| Temp ( °C) | 0.29 | 0.20 | 0.27 | 0.29 | 0.30 | |
| RH (%) | 0.39 | 0.20 | 0.26 | 0.45 | 0.34 | |
| Chilbolton | FL | CL1 | CL2 | CL3 | CL4 | |
| Temp ( °C) | 0.12 | 0.11 | 0.13 | 0.15 | 0.12 | |
| RH (%) | 0.22 | 0.14 | 0.20 | 0.19 | 0.18 | |
| Capel Dewi | FL | CL1 | CL2 | CL3 | CL4 | CL5 |
| Temp ( °C) | 0.05 | 0.07 | 0.07 | 0.05 | 0.09 | 0.06 |
| RH (%) | 0.08 | 0.05 | 0.06 | 0.19 | 0.18 | 0.06 |
| Davidstow | FL | CL1 | CL2 | CL3 | CL4 | CL5 |
| Temp ( °C) | 0.38 | 0.05 | 0.19 | 0.53 | 0.18 | 0.41 |
| RH (%) | 0.31 | 0.06 | 0.16 | 0.51 | 0.31 | 0.42 |

been found to have higher bacterial emissions (Shaffer and Lighthart, 1997). Cluster 3 originates locally, with an additional source from a south-westerly wind direction at wind speeds <6 m s⁻¹, and similar to Cluster 5 and Cluster 4, a source from a NNW wind direction. The source of Cluster 3 may originate from the woodland or the improved grassland around the site, but comprise wet discharged fungal spores. Similarly, Cluster 1 and Cluster 2 were suspected to be wet discharged fungal spores, each of which show a similar trend and indicate a source to the east, at low wind speeds, which is a predominantly improved grassland area.

### 3.5 Statistical relationship between fluorescent particles per site and meteorological data

Third-order polynomial fits were fitted to the whole data to identify relationships between the total fluorescent particles and clusters per site, and their relationship to the meteorological variables, relative humidity and temperature. The R values per polynomial fit are presented in Table 5. The relationships between the variables are reasonably low, and do not always present any strong or moderate R values (with the exception of Davidstow Cluster 3 and Cluster 5). In comparison, the correlation coefficient between fluorescent particles and meteorological data when averaged over the entire campaign produced more apparent trends for data collected in Munnar, in southern India by Valsan et al. (2016).

### 3.6 Sensitivity of HAC to fluorescent threshold

The data from each site were initially analysed using the original recommendation of a baseline forced trigger (FT) ± 3SD from Kaye et al. (2005) as based on a limited laboratory data set, prior to the use of FT ± 9SD. Chilbolton produced a four-cluster solution when analysed using FT ± 9SD, and produced a five-cluster solution when analysed using FT ± 3SD. Concentrations, especially for Cluster 3 (9SD), are lower than the original 3SD cluster (which for 3SD bar plot, is Cluster 4) (Fig. 7). Fluorescent




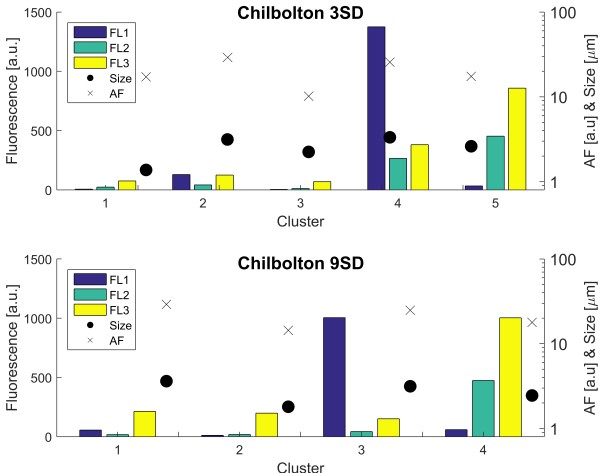

**Figure 7.** Sensitivity of the HAC threshold using 3SD and 9SD for analysis of the Chilbolton dataset

particle number decreases when using the 9SD threshold, consisting of 427,642 fluorescent particles compared to 1,427,843 fluorescent particles when using 3SD. When adopting the 9SD threshold potential interferents were removed, which represents a more robust solution for biological particle detection, although low activity within aged biological particles may mean these are removed as well.

## 4  Conclusions

Hierachical agglomerative cluster (HAC) analysis was used to characterise biological particles from four different sites in the United Kingdom. Of these sites, three were located in a rural environment (Davidstow, Chilbolton, and Capel Dewi), and one located on the coast of England (Weybourne). Using HAC provided a four-cluster solution for both datasets collected using a WIBS-3, and a five-cluster solution for the two datasets collected using a WIBS-4. Similarities in cluster properties could be identified between sites when using the same version of instrument (e.g. Chilbolton and Weybourne).

The use of HAC produced distinct fluorescent signatures for each cluster. Often, clusters which displayed a similar fluorescence profile were segregated due to differences in particle size and shape, a phenomena which occurred for most sites. Though most clusters could be attributed as a certain biological particle type, Weybourne Cluster 1 was unique, and was deemed to be a mixture of two different particles, owing to different activity compared to other clusters and potentially two sources of particles as identified using the wind speed and direction with the LCM2015.

The use of forced trigger ± 9SD for baseline fluorescence, as opposed to ± 3SD, resulted in the loss of a cluster for the Chilbolton dataset (as illustrated in Section 3.6). However, for all sites, the clusters were considered to be biological following analysis of the meteorological data, which may result from the use of FT ± 9SD, in which potential interferents were removed.



There were difficulties when using HAC and the fluorescent signal intensities for particle identification, and using this alone would, for the most part, provide incorrect particle identification (Table 4). For example, Weybourne Clusters 1 and 3, exhibited high fluorescence in channel FL3, which was reminiscent of bacteria (Dstl Experiment 2014). However, once the abundance of these clusters and the location of the site were considered, it was deemed unlikely that these were bacterial clusters.

When using the meteorological data, the initial assumptions made were either discredited or improved upon. For the clusters that were considered to be interferents, owing to the high signal in channel FL2, meteorological data analysis disproved this, and indicated clusters of biological origin. Similarly, the clusters which comprised Group 5 were considered to be pollen fragments, owing to the high FL3 signal. There were differences of high (Davidstow Cluster 1) to moderate (Chilbolton and Davidstow Cluster 4) signal in channel FL2, and it was speculated that the dominance of channel FL2, when accompanied by

a high signal in channel FL3, is what separates pollens from fungal spores. However, following analysis of the meteorological data, the Group was determined to comprise wet discharged fungal spores (Chilbolton and Weybourne Cluster 4), and bacteria (Davidstow Cluster 1). Though the original identity inferred using the fluorescent profile was incorrect, the dominance of the FL2 signal provided some distinction between bacteria from fungal spores.

Between sites, the most common cluster class was wet discharged fungal spores, and for Weybourne, Chilbolton, and Capel

Dewi, they were the most abundant particle class, comprising nearly all the clusters at each site. In contrary, Davidstow, a site comprising mostly improved grassland, with some arable and horticultural land nearby, comprised mostly bacterial clusters. The Davidstow site is located close to a dairy factory, and though the polarplots do not directly show any emissions from such, it is possible that there is some contribution from here, as sites with similar land cover properties do not experience similar high concentrations.

The use of land cover maps in relation to polarplots proved useful for identifying the sources of clusters. The identity of Weybourne Cluster 1 was unconfirmed following analysis of the cluster diurnal variation and meteorological data. However, using the LCM2015 in relation to the polarplot inferred that there were two potential sources of Weybourne Cluster 1, originating from two differing land types, providing evidence that the cluster comprised a mixture of two particles. Certain cluster types could not be clearly attributed to a specific land cover type, however, more wet discharged fungal spores were found to

be associated with improved grassland areas. For clusters which were inferred to be bacteria, potential sources ranged from broadleaf and coniferous woodland (Capel Dewi Cluster 4) to a mixture of potential sources including urban, arable, and improved grassland (Davidstow Cluster 2, 3, and 5).

In general, the source of the total fluorescent particles at Chilbolton originated locally, from arable and horticultural land, with some contribution from the mushroom farm located SW of the site, and were inferred to comprise wet discharged fungal

spores. At Davidstow, total fluorescent particles originated from a WSW wind direction at wind speeds of 5 -10 m s-1, covering mostly improved grassland but also some urban and suburban land. The identity of these particles were assumed to be bacteria, potentially originating from the urban and suburban land, but more likely influenced by emissions from the dairy factory located north-west of the site. At Weybourne it was assumed that the wet discharged fungal spores originated from mostly improved grassland owing to the lower wind speed. At Capel Dewi, total fluorescent particles originate locally, and also from a NNW

wind direction at wind speeds >8 m s-1. The difficulties in assigning a particle type once analysing the diurnal variation and





meteorological data may result from the fact that there are two sources at Capel Dewi, with suspected bacteria originating from the broadleaf and coniferous woodland from the NNW and wet discharged fungal spores originating locally from the improved grassland.

To our knowledge this is the first use of both ArcGIS land cover mapping, in association with airborne bioaerosol concen-
5   trations, to identify distinctive emission patterns and factors. This analysis relied upon the use of HAC, and as one of multiple unsupervised clustering approaches, it is possible that this analysis is subject to various levels of misclassification depending on the instrument used. Whilst the ancillary ambient data included in this study supports some of the derived cluster variability, it is not possible to comment further on this issue without additional laboratory data on known bioaerosol types. This is the subject of ongoing work and will be assessed in the near future, with the discussion of results from the referenced 'Dstl
10   experiment 2017' to be published later this year.

For future studies, more knowledge of the reaction of speciated biological particles to differences in meteorology, especially relative humidity and temperature would aid characterisation in studies such as this.





# Appendix A

**A1**

*Competing interests.* The authors declare no competing interests

*Acknowledgements.* The lead author is funded under the DSTL (Defence Science and Technology Laboratory) and DGA (Direction Générale
de l'Armement) Anglo-French PhD scheme and affiliated to the NERC EAO Doctoral Training Partnership (ORCID iD: 0000-0003-0719-3258). Data from Chilbolton were collected during the NERC APPRAISE programme (Aerosol Properties, PRocesses And Influences on the Earth's Climate), and from Davidstow during the the COnvective Precipitation Experiment (COPE).




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
