# Peer review of "Characterisation and source identification of biofluorescent aerosol emissions over winter and summer periods in the United Kingdom"

_Atmospheric Chemistry and Physics, 2018_

## Referee Comment (RC1) · Anonymous Referee #1 · 12 Sep 2018

General comments

In this paper, the authors report on the deployment of WIBS-3 and WIBS-4 sensors in four ground sites in the U.K. The collected dataset is very extensive, covering different locations and seasons. Records like these of fluorescent particle concentrations are of current interest in the community.

However, I found some of the work on the HAC clustering and particle identification to be speculative at times. I think it could be made more convincing with the use of laboratory data that the authors reference, but don't quite show. From what I understand, the cluster types were initially assigned to HAC-derived clusters following broad

observations of similarity to laboratory types. I think the analysis would be significantly strengthened if a direct statistical comparison of lab and field clusters were presented. Since HAC method validation is a major part of this paper, I also feel that presenting clustering quality metrics would help.

I recommend the publication of this paper in ACP, after the following major comments are addressed:

(1) Can you discuss further how the clusters are initially assigned to types following laboratory work? Has the proximity of lab-derived clusters to field clusters been assessed or calculated? Can you use a distance metric to directly compare them? Basically, how sure are you of the assignments of the field-derived clusters to cluster types shown in Table 2?

(2) Similarly, an inter-cluster distance metric could be used to support the segregation of initial clusters into distinct groups in Table 3. Maybe using a cluster dendrogram plot to show similarity would be a good idea?

(3) Can you present any metrics on how successful the HAC algorithm was in segregating particles into different clusters? How distinct are the clusters? Consider presenting criteria such as Calinski-Harabasz index or Davies-Bouldin index to demonstrate the cluster separation quality.

(4) There is a discussion in section 3.6 that shows that assigning a different fluorescence threshold caused a completely different clustering solution for the Chilbolton dataset. Is the extra cluster found with the 3SD threshold comprised solely of interferents? I would be interested to see more discussion of how this affects the other datasets.

(5) More discussion of possible interferents would help. What do you think they are? How do they compare to previous laboratory studies?

Specific comments

Abstract, pg. 1 line 18 (last sentence): consider rephrasing, not sure what this means.

Introduction, pg. 3 line 28: principal should be principle

Introduction, pg. 5 line 5: can you discuss thresholding here briefly? Why was a different threshold used in this work? What are the advantages?

Methods, pg. 5, line 16: this sentence (starting with "Whilst...") seems unfinished.

Section 3.4.1: Temperature and Relative Humidity: consider providing more figures for this analysis. It would be useful to see if all temperature and RH trends for clusters identified as fungi vs. bacteria match each other. It is much harder to see from just a text description.

Section 3.4.2: Wind Speed and Wind Direction: similarly, would it be possible to show wind roses of each cluster (or cluster group) to make the similarities discussed in text more obvious?

Section 3.5: Statistical relationship between fluorescent particles per site and meteorological data: what is the purpose of this analysis? In particular, why were third-order polynomial fits used? Given that the statistical model is not fully clear and that all of the r-squared values are low, consider either significantly expanding this section or eliminating it.

---

## Referee Comment (RC2) · Anonymous Referee #2 · 26 Sep 2018

General comments.

The manscript concerns the observations of bioaerosols using WIBS-3 and WIBS-4 and their main classification into types of bioaerosols using clustering, their typical patterns and potential sources, where the source analysis has been done with ArcGIS. The observational period cover 4 sites in the UK and the observations are Jan-March 2009, June-August 2013 (WIBS-4), Feb-June 2013 (WIBS-4) and August 2009 (WIBS-3). Please find below a numbered set of comments to the manuscript as well as one specific comment.

1. The manuscript cover an area there is of relevance to ACP and an area where there

is very few studies.

2. The study itself contains a large data set, about 9 months of data of bioaerosols obtained with WIBS instruments. However the data itself are not part of the manuscript, but only coarse numerical summaries.

3. The conclusions in the manuscript are mainly related to clustering of data into 4 or 5 main clusters and there are some indication to potential source areas. These conclusions seem however to be indicative and qualitative instead of quantitative.

4. The scientific methods are valid and clearly described

5. The results and the methods are not described in depth to reach the current conclusion. This relates to both clustering and the mapping using ArcGIS. See issues below

5a. Issues on mapping : There is no exact geographical location of the sites. Please add this to the manuscript

5b. Issues on mapping: I could identify the Weybourne observatory and compared Figure 1 with both google maps and land cover 2015 (Digimap). The land cover in Fig 1 shows large amounts of Coniferous woodland near Weyborne. However Googlemaps and Digimap shows that this area is improved grassland. Is this a simple mapping mistake when drawing figure 1 or is there a more systematic mistake in the manuscript, where the land cover has not been used correctly for all the sites?

5c. Issues on mapping. Several times in the manuscript including the conclusion there is a connection between the observations and specific farming activities. However the manuscript does not contain any information about farm location. This connection can therefore not be made unless such data are present. Furthermore, why have those specific farms been attributed as source and not other farms in the area?

5d. Issues on mapping. The chosen land cover map is probably among the best maps in the UK. However it has some limitations. Smaller features such as smaller

woodlands are not part of this map. The authors have not taken these limitations into account.

5e. Issues on clustering. The clustering use an approach by Crawford et al. (2015). This requires use of dry materials that are aerosolised and added to the instrument in a laboratory. This calibration data is not present in this paper.

5f. Issues on clustering. The paper by Crawford et al. (2015) only describe pollen but not if other bioaerosols have been used. Crawford et al. writes that the four pollen types are common in the UK. This is not correct. Two of the four allergens (paper mulberry and ragweed) are rare in the UK. The third in Crawford (birch) is common in the UK, typically with a season in April. This suggest that in this manuscript only Capel Dewi would have had a chance to detect this. The fourth pollen in Crawford et al (2015) is ryegrass. However, the pollen size is typically 30-40, which is above the typical detection limit of the WIBS. Have the authors also calibrated with pollen and have they also used pollen that are less likely to be in the UK atmosphere and less likely to be detected by the WIBS?

5g. Issues on clustering. In the paper by Crawford et al (2015), the team has used dry pollen. Dry pollen from commercial samples will have a very different shape to fresh airborne pollen as pollen can take up and loose water. Using dry pollen will generally cause poor calibration of real-time instruments as the shape of dry pollen is very different compared to fresh pollen. Secondly has there been any investigations if dry pollen will cause different excitation compared to fresh pollen

6. The methods section are generally good if the issues in section 5 can be solved

7. The citations and reference list seems to be up-to-date with a good selections of citations to new and relevant literature. However the manuscript is not clear where the studies confirms existing knowledge and more importantly where it contributes with new knowledge by positioning the results against published literature

8. The title of the paper reflects parts of the study, but not the part that try to associate the observed bioaerosols with potential sources (the ArcGIS part)

9. The abstract cover well the contents of the paper.

10. The presentation is generally clear and well structured but the conclusion might need some work (see point 13)

11. The language is generally clear and fluent and does not need further improvements

12. 12. The manuscript does not include mathematical formula. However the manuscript decribes the use of a third order polynomia with R values between the observations and the polynomia (Table 5). The polynomia is not found anywhere in the manuscript (or in supplementary information) and the results (including low R values) will probably need a discussion.

13. The conclusion is almost two pages and part of the conclusion seems to be a discussion (e.g. the section concerning difficulties in the clustering). Maybe the conclusion should be shortened to make it more sharp and part of the material should be moved to the discussion section. If the authors have used calibration of the instrument against known material, then this calibration needs to be described in more detail and in particular how well the instrument is able to identify test samples similar to the calibration material.

14. There seem to be 60-65 references in the manuscript. This seems appropriate for this type of manuscript

15. There is no supplementary information. The authors might consider if adding supplementary information can improve transparency of the methods and the documentation.

Specific comments

On page 25, line 4 onwards, the authors write that this is the first time ArcGIS has been

used in relation to land cover mapping and bioaerosols to derive emission patterns etc. As far as I know there are many such studies (some of them are in fact in the reference list), but it is the first time it has been done in connection with the WIBS instrument.

---

## Author Comment (AC1) · 16 Nov 2018

**RESPONSES TO ANONYMOUS REFEREE #1**

(1) Reviewer comments are in black text.
(2) Author responses are in blue text.
**(3) Additions/modifications made to the manuscript.**

General comments
In this paper, the authors report on the deployment of WIBS-3 and WIBS-4 sensors in four ground sites in the U.K. The collected dataset is very extensive, covering different locations and seasons. Records like these of fluorescent particle concentrations are of current interest in the community.

However, I found some of the work on the HAC clustering and particle identification to be speculative at times. I think it could be made more convincing with the use of laboratory data that the authors reference, but don't quite show. From what I under-stand, the cluster types were initially assigned to HAC-derived clusters following broad observations of similarity to laboratory types. I think the analysis would be significantly strengthened if a direct statistical comparison of lab and field clusters were presented. Since HAC method validation is a major part of this paper, I also feel that presenting clustering quality metrics would help.

I recommend the publication of this paper in ACP, after the following major comments are addressed:

We thank the reviewer for this comment and, in the following responses, clarify why this will be the subject of future work. This work utilised data and analysis from published studies (e.g. Savage et al 2017, Hernandez et al 2016) and whilst some of the other laboratory data used in this study needs to be subject to the peer review process, we felt alluding to recent results, which we will publish imminently, would help the classification process.

(1) Can you discuss further how the clusters are initially assigned to types following laboratory work?

The fluorescent signals of ambient derived clusters have been compared to laboratory data using trends in fluorescent channels in order to initially group the 18 clusters for further analysis. Specifically, the clusters from each field site were compared to existing ('Dstl experiment 2014') and new ('Dstl experiment 2017') laboratory data, depending upon the instrument used, in addition to published data e.g. Savage et al 2017. Laboratory data was available from a WIBS-3 for the 2014 Dstl dataset, and from a WIBS-4 for the 2017 Dstl dataset (the results from such are to be published in the new year). Prior to comparing the ambient clusters to the laboratory data according to broad fluorescent signature, the process for deriving these clusters was the same as used in all previous ambient studies. Additionally, data from published laboratory experiments (e.g. Savage et al 2017, Hernandez et al 2016) were used to provide some further support, and aid the initial grouping of these clusters into suspected particle type groups.

Has the proximity of lab-derived clusters to field clusters been assessed or calculated?

These have not been calculated in this paper directly; rather a qualitative comparison made between trends across fluorescent channels, size and shape. To perform this in a quantitate manner requires consideration of a number of issues which require further laboratory data to be published and subject to the peer review process. Firstly, rather than using unsupervised methods [in this case hierarchical clustering, HAC], supervised techniques would be able to assign any sampled ambient particle to a class that has been studied in the laboratory depending on choice of parameters use for any given technique. Part of this procedure includes choice of appropriate distance metric between each fluorescence signal [which the reviewer refers to as a proximity metric]. These methods demonstrate exciting potential for improved and more detailed bio-aerosol classification. However, as noted in Ruske et al (2018), before recommendations can be given to choice of method and distance metric, more laboratory data is needed to reduce the chance of misclassification. Indeed, even for HAC, Ruske et al 2018 studied a range of model permutations, demonstrating the variability in laboratory signatures according to how samples were prepared, for example. We are planning on a conducting a much more thorough evaluation of statistical methods once we have published and had this data appropriately peer reviewed. In this paper we use the current recommended configuration HAC as used in all current bio-aerosol publications.

Can you use a distance metric to directly compare them?

Please see our response to the previous questions. It is entirely possible to employ a distance metric to directly compare ambient clusters and laboratory data; this information would be explicitly used within supervised learning techniques to perform direct classification. However we feel it would serve no real benefit to detail those distances without then using the supervised techniques. Indeed, this goes beyond the scope of this particular piece of work and is inline the current state of the literature. The idea behind the use of both laboratory data and published data in this study was to qualitatively compare the fluorescent profiles of known biological types to ambient data to group these clusters into suspected particle types for further analysis. This is similar to the approach taken by Kasparian et al. (2017).

Basically, how sure are you of the assignments of the field-derived clusters to cluster types shown in Table 2?

There is undoubtedly a level of error with this method, not least by qualitatively using laboratory data, which may not be fully representative of 'real-world' conditions i.e. not accounting for the effects of atmospheric transport, aggregation and fragmentation of particles. However, the use of such a method has been employed previously to determine potential cluster particle types (e.g. Crawford et al 2017, Kasparian et al. 2017) and is still a valuable method for sanity checking profiles.

Here, we did not rely only on the expected fluorescent signals from laboratory data to determine the type of particles these clusters comprise. Instead, we built upon this by considering abundance, the size and shape of the particles within the cluster, the diurnal variation of the cluster, the response to the meteorological variables (temperature and relative humidity), and the land cover category for the site location in question. The assignments following this process (Table 4 in the manuscript) either reinforces the initial assumptions made using only the laboratory data, or disproves it at each stage. A similar approach was used by Crawford et al., (2014) where the identified PBAP clusters were found to correlate well with other bioaerosol detection techniques (Gosselin et al., 2016).

(2) Similarly, an inter-cluster distance metric could be used to support the segregation of initial clusters into distinct groups in Table 3. Maybe using a cluster dendrogram plot to show similarity would be a good idea?

The use of the Calinski-Harabasz index to segregate the clusters has been employed in this paper, similar to previous studies (e.g. Crawford et al 2017). The segregation of clusters into the six groups as seen in Table 3 are based on the fluorescent profile analysis and comparison to laboratory data between the four sites as conducted in section 3.2.

An example cluster dendrogram can be seen for Weybourne (Fig.1), which is accompanied by the Weybourne centroid figure. Using the cluster dendrogram plot it can be seen that the merging of Cluster 3 and Cluster 2 occurs first, which is interesting considering the differences in fluorescent signal which can be seen in the centroid figure. As Cluster 3 was the dominant cluster at this site, the merge with Cluster 1 is not unexpected, owing to the similar fluorescent profile between the two clusters.  The clusters then merge with Cluster 4 last which has a slightly similar signal to Cluster 1 and Cluster 3 in terms of the higher FL3 signal, but differs with some signal in FL2 and FL1.

Though this illustrates the process of merging the clusters, this does not have the advantage of showing the characteristics of the clusters, such as the particle size and shape, compared to the centroid figures used in this study. A more detailed discussion of the application, interpretation and limitation of the Calinski–Harabasz index applied to these instruments may be found in the related publication by Ruske et al. 2018. Additionally, a more detailed description of HAC clustering using dendrograms is described in Ruske et al. (2018). The hierarchy using the original strategy suggested in Crawford et al. (2015) compared with the modification using a 9 sigma threshold as suggested by Savage et al. (2017) is also discussed.

[Figure]

*Figure 1: Example cluster dendrogram for Weybourne in addition to the centroid figure used in this study.*

(3) Can you present any metrics on how successful the HAC algorithm was in segregating particles into different clusters?
How distinct are the clusters?
Consider presenting criteria such as Calinski-Harabasz index or Davies-Bouldin index to demonstrate the cluster separation quality.

As shown in Figure 2, when using the Calinski-Harabasz (CH) criterion for segregating the clusters for Weybourne, the optimum value suggests a four-cluster solution for the data from this site. As highlighted in the paper, it is common that similar clusters are often subsets, segregated by particle size and shape.

[Figure]

*Figure 2: Example Calinski-Harabasz cluster solution following clustering of Weybourne.*

(4) There is a discussion in section 3.6 that shows that assigning a different fluorescence threshold caused a completely different clustering solution for the Chilbolton dataset. Is the extra cluster found with the 3SD threshold comprised solely of interferents? I would be interested to see more discussion of how this affects the other datasets.

When using the 3SD threshold for the Chilbolton site, we note that the change in threshold does not result in a completely different clustering solution (Figure 3). Rather, it can be seen when using 9SD that Cluster 3 and 4 are representative of Cluster 4 and 5 when using 3SD, but at lower concentrations. Considering the presence of some signal in channel FL1, but a dominant signal in channel FL3, it can be assumed that Cluster 2 (3SD) is representative of Cluster 1 (9SD). This leaves Clusters 1 and 3 (3SD), which may have been merged as they appear to be represented by Cluster 2 (9SD). This extra cluster was determined to be a wet-discharged fungal spore following the complete analysis as opposed to interferent particles.

[Figure]

*Figure 3: Comparison figure showing the difference between 3SD and 9SD for Chilbolton (Figure 7 in manuscript).*

At the other sites there is not always a loss of a cluster as in Chilbolton. In comparison, some sites retained the same number of clusters (e.g. Davidstow and Weybourne), whilst the change from 3SD to 9SD for Capel Dewi resulted in the gain of a cluster, from four clusters when using 3SD to five clusters when using 9SD.

**As a result of challenges in interpreting cluster solutions when using FT + 3SD and 9SD at each site, these plots been added to the supplementary materials (on pages 16-17).**

(5) More discussion of possible interferents would help.
What do you think they are?
How do they compare to previous laboratory studies?

The four sites in this paper are similar in that they are not closely located to any major cities or towns and are similarly situated in agricultural/grassland locations. This reduces the potential impacts of vehicle emissions from city traffic and fuel burning and other sources, but does not rule out some episodic emissions from roads or access points located close to a few of the sites.

In the Hernandez et al 2016 laboratory study  the dominance of the FL2 channel (referred to as Type B in their study following ABC analysis)  was determined  to be representative of potential interferents . By using a higher 9SD threshold for analysis and comparing this to the 3SD the fluorescent signal intensity decreases, even when, for some sites, the number of clusters does not change. For Capel Dewi, the amount of clusters increases, while Weybourne and Davidstow stay the same.

Though, given that there are no loss of clusters for the other sites, and even the production of an additional cluster for Capel Dewi, the use of 9SD here produces only a reduction in the fluorescent fraction of each cluster. The use of 9SD for Capel Dewi appears to split Cluster 4 (3SD) into two different clusters (Cluster 1 and Cluster 2) when using the 9SD threshold.

SPECIFIC COMMENTS:

Abstract, pg. 1 line 18 (last sentence): consider rephrasing, not sure what this means.

This sentence is a comment on the lack of available published information of different particle species and the influence of meteorological variables (such as RH and Temp) on their abundance.

This has been changed on page 1 and page 25 from:
*"More knowledge of the reaction of speciated biological particles to differences in meteorology, such as relative humidity and temperature would aid characterisation studies such as this."*

To:
***"More published data and information on the reaction of different speciated biological particles types to fluctuations in meteorological conditions, such as relative humidity and temperature, would aid particle type characterisation in studies such as this."***

Introduction, pg. 3 line 28: principal should be principle

**We have changed principal to principle on page 3.**

Introduction, pg. 5 line 5: can you discuss thresholding here briefly? Why was a different threshold used in this work? What are the advantages?

**A brief description has been added to page 5 to describe the use of the 9SD threshold.**
*"Contrary to previous work, this is additionally the first use of a differing fluorescent threshold of 9 standard deviations (SD) compared to traditionally 3SD, in an ambient setting,* **to reduce the impact of interferents from potential anthropogenic sources, following Savage et al 2017."**

Methods, pg. 5, line 16: this sentence (starting with "Whilst . . .") seems unfinished.

The sentence in this paragraph on page 5 has been re-worded from:

*"Whilst Skjøth et al. (2012) utilised the Corine Land Cover 2000 dataset to identify agricultural areas under rotation and in harvest in relation to Alternaria spore concentrations in Denmark."*

To:

*"In addition, Skjøth et al. (2012) utilised the Corine Land Cover 2000 dataset to identify agricultural areas under rotation and in harvest in relation to Alternaria spore concentrations in Denmark"*

Section 3.4.1: Temperature and Relative Humidity: consider providing more figures for this analysis. It would be useful to see if all temperature and RH trends for clusters identified as fungi vs. bacteria match each other. It is much harder to see from just a text description.

The authors acknowledge that inclusion of these plots would aid interpretation of the paper, as opposed to a text-based description. We were keen to include these plots in the manuscript, however owing to the quantity of plots, these would take up a considerable amount of space. **We have added the suggested figures, showing the differences observed in relation to temperature and relative humidity, to the supplementary materials (pp. 2 – 9).**

Section 3.4.2: Wind Speed and Wind Direction: similarly, would it be possible to show wind roses of each cluster (or cluster group) to make the similarities discussed in text more obvious?

Due to the amount of clusters including these figures in the manuscript would consume a considerable amount of the paper, which is the reason why only the total fluorescent polar plots from each site were included. **As a result these plots have been added to the supplementary materials (pp. 12 – 15).**

Section 3.5: Statistical relationship between fluorescent particles per site and meteorological data: what is the purpose of this analysis? In particular, why were third-order polynomial fits used? Given that the statistical model is not fully clear and that all of the r-squared values are low, consider either significantly expanding this section or eliminating it.

The purpose of this analysis was to produce a value to be used to calculate an emission factor, similar to Crawford et al (2014), due to the sparsity of data relating to bioaerosol emission and various meteorological drivers. Due to the variance in the total fluorescent data from each site and the cluster variability third-order polynomial fits were chosen. **As a result of the already lengthy analysis in the manuscript, this section has been removed.**

**References**

Crawford, I., Robinson, N. H., Flynn, M. J., Foot, V. E., Gallagher, M. W., Huffman, J. A., Stanley, W. R., and Kaye, P. H.: Characterisation of bioaerosol emissions from a Colorado pine forest: results from the BEACHON-RoMBAS experiment, Atmos. Chem. Phys., 14, 8559-8578, https://doi.org/10.5194/acp-14-8559-2014, 2014.

Crawford, I., Gallagher, M. W., Bower, K. N., Choularton, T. W., Flynn, M. J., Ruske, S., Listowski, C., Brough, N., Lachlan-Cope, T.,  Flemming, Z. L., Foot, V. E., and Stanley, W. R.: Real Time Detection of Airborne Bioparticles in Antarctica, Atmospheric Chemistry and Physics Discussions, pp. 1–21, https://doi.org/10.5194/acp-2017-421, http://www.atmos-chem-phys-discuss.net/acp-2017-421/, 2017.

Gosselin, M. et al., 2016. Fluorescent bioaerosol particle, molecular tracer, and fungal spore concentrations during dry and rainy periods in a semi-arid forest. *Atmospheric Chemistry and Physics*, 16(23), pp.15165–15184.

Hernandez, M., Perring, A. E., McCabe, K., Kok, G., Granger, G., and Baumgardner, D.: Chamber catalogues of optical and fluorescent signatures distinguish bioaerosol classes, Atmospheric Measurement Techniques, 9, 3283–3292, https://doi.org/10.5194/amt-9-3283-2016, 2016.

Kasparian, J. *et al*. Assessing the Dynamics of Organic Aerosols over the North Atlantic Ocean. *Sci. Rep.* **7**, 45476; doi: 10.1038/srep45476 (2017).

Ruske, S., Topping, D. O., Foot, V. E., Morse, A. P., and Gallagher, M. W.: Machine learning for improved data analysis of biological aerosol using the WIBS, Atmos. Meas. Tech. Discuss., https://doi.org/10.5194/amt-2018-126, in review, 2018.

Savage, N., Krentz, C., Könemann, T., Han, T. T., Mainelis, G., Pöhlker, C., and Huffman, J. A.: Systematic Characterization and Fluo- rescence Threshold Strategies for the Wideband Integrated Bioaerosol Sensor (WIBS) Using Size-Resolved Biological and Interfering Particles, Atmospheric Measurement Techniques, pp. 1–41, 2017.

---

## Author Comment (AC2) · 16 Nov 2018

**RESPONSES TO ANONYMOUS REFEREE #2**

(1) Reviewer comments are in black text.
(2) Author responses are in blue text.
**(3) Additions/modifications made to the manuscript.**

We thank the reviewer for their detailed comments and helpful recommendations, which are addressed in the following responses.

The manuscript concerns the observations of bioaerosols using WIBS-3 and WIBS-4 and their main classification into types of bioaerosols using clustering, their typical patterns and potential sources, where the source analysis has been done with ArcGIS. The observational period cover 4 sites in the UK and the observations are Jan-March 2009, June-August 2013 (WIBS-4), Feb-June 2013 (WIBS-4) and August 2009 (WIBS- 3). Please find below a numbered set of comments to the manuscript as well as one specific comment

1. The manuscript cover an area there is of relevance to ACP and an area where there is very few studies.

2. The study itself contains a large data set, about 9 months of data of bioaerosols obtained with WIBS instruments. However the data itself are not part of the manuscript, but only coarse numerical summaries.

We would like to re-affirm that we will provide access to the raw data as per Copernicus data sharing guidelines, but focus on the scientific elements of our analysis in this paper.

3. The conclusions in the manuscript are mainly related to clustering of data into 4 or 5 main clusters and there are some indication to potential source areas. These conclusions seem however to be indicative and qualitative instead of quantitative.

We would like to clarify that the cluster procedure provides and quantitative analysis of the UV-LIF spectral data but we have qualitatively assigned these clusters to potential bio-aerosol types using a standard comparison of fluorescence profiles. As we note in response to reviewer 1, the fluorescent signals of ambient derived clusters have been compared to laboratory data using trends in fluorescent channels in order to initially group the 18 clusters for further analysis. Specifically, the clusters from each field site were compared to existing ('Dstl experiment 2014') and new ('Dstl experiment 2017') laboratory data, depending upon the instrument used, in addition to published data e.g. Savage et al 2017. Laboratory data was available from a WIBS-3 for the 2014 Dstl dataset, and from a WIBS-4 for the 2017 Dstl dataset (the results from such are to be published in 2019). Prior to comparing the ambient clusters to the laboratory data according to broad fluorescent signature, the process for deriving these clusters was the same as used in all previous ambient studies. Additionally, data from published laboratory experiments (e.g. Savage et al 2017, Hernandez et al

2016) were used to provide some further support, and aid the initial grouping of these clusters into suspected particle type groups.

4. The scientific methods are valid and clearly described

5. The results and the methods are not described in depth to reach the current conclusion. This relates to both clustering and the mapping using ArcGIS. See issues below

We apologise if the reviewer feels this is the case, and we hope our detailed responses given here and the changes to the manuscript address this point.

5a. Issues on mapping: There is no exact geographical location of the sites. Please add this to the manuscript

**We have added the geographic location of each site to Table 1 in section 2.3.**

5b. Issues on mapping: I could identify the Weybourne observatory and compared Figure 1 with both google maps and land cover 2015 (Digimap). The land cover in Fig 1 shows large amounts of Coniferous woodland near Weyborne. However Googlemaps and Digimap shows that this area is improved grassland. Is this a simple mapping mistake when drawing figure 1 or is there a more systematic mistake in the manuscript, where the land cover has not been used correctly for all the sites?

It is correct that there is no coniferous woodland around the Weybourne site, however, the land cover map used shows that the area around the Weybourne site comprises improved grassland, not coniferous woodland. This may be unclear owing to the two shades of green used to identify improved grassland areas and coniferous woodland. **To resolve this, a clearer distinction has been made between the two shades of green, with a much lighter green colour to represent improved grassland, and a darker shade of green for the coniferous woodland (Figure 1).**

[Figure]

*Figure 1 – (Left) Weybourne LCM and colour bar from the previous manuscript, (Right) a zoomed out Weybourne map to illustrate the new colour bar with distinction between coniferous woodland (seen in the bottom right) and improved grassland (around sample site).*

5c. Issues on mapping. Several times in the manuscript including the conclusion there is a connection between the observations and specific farming activities. However the manuscript does not contain any information about farm location. This connection can therefore not be made unless such data are present. Furthermore, why have those specific farms been attributed as source and not other farms in the area?

There are particular sampling sites which are in close proximity to a dairy factory (Davidstow) and a mushroom/composting facility (Chilbolton) which were observed during each experimental campaign within narrow wind sectors. Due to the nature of the land cover maps, details such as farm locations/nearby areas of interest are not included. As such, there was some consideration given to include a separate map of each site (either a simple OS basemap or Satellite imagery) outlining any potential influencing sources. However, adding such extra maps would add substantially to the size of the manuscript. As an alternative, altering the transparency of the land cover map and overlaying this upon an OS basemap has allowed for further information of the surrounding area at each site to be presented (Figure 2).

Additionally, 'Potential point sources' have been added to the maps from each site and have been split into 'Farming sites' and 'Other potential influencing sources' as identified during each campaign and with the use of online maps (e.g. Google Maps). Apart from the dairy factory at Davidstow, this site now features other potential point sources, including a dairy farm and other surrounding farms, a small garden centre, and a slaughterhouse which are in close proximity to the sampling site. Additionally, the presence of a livestock breeder near to the MST Capel Dewi site has been added. A description of these influencing sources has been added to the text.

**With regards to the changes made in the manuscript, the land cover maps from each site have been overlaid on OS basemaps to provide geographical context to each site location. Additionally, potential influencing sources have been added to illustrate farming activity or any other potential sources of biological or interferent particle material, which can be seen in Figure 2 (Figure 1 in the manuscript).**

**A sentence has been added to Section 2.1, paragraph 3, to explain this change -**
*"To provide geographic context to each site, the LCM2015 has been overlaid on an OS basemap. Additionally, the presence of local farming activity and other potential influencing sources are illustrated in Figure 1"*

[Figure]

*Figure 2 – LCM layer overlaid on OS basemaps, potential point sources are split into farming sites and 'others', illustrated by black/white asterisks respectively.*

5d. Issues on mapping. The chosen land cover map is probably among the best maps in the UK. However it has some limitations. Smaller features such as smaller woodlands are not part of this map. The authors have not taken these limitations into account.

The chosen land cover map does have its limitations; however, the advantages do outweigh these, especially as this is an up-to-date land cover map. The method in which the land cover map is produced does mean that smaller features are excluded from the map, which is noticeable when comparing a basemap to the LCM. A sentence will be added to the manuscript to ascertain that there may be some smaller features which have been excluded.

**The statement below has been added to Section 2.1, paragraph 3.**

*"Though the LCM2015 provides up-to-date data on the land cover characteristics of each site, it is acknowledged that smaller features are not identified and thereby not considered as potential sources."*

5e. Issues on clustering. The clustering uses an approach by Crawford et al. (2015). This requires use of dry materials that are aerosolised and added to the instrument in a laboratory. This calibration data is not present in this paper.

We refer the readers to Crawford et al (2015) and noted data access procedures in the paper for the calibration data used in that study. Whilst the use of hierarchical agglomerative clustering (HAC) correctly attributed 98% of laboratory generated fluorescent test particle data, its limitation on more representative biological aerosol is noted in our latest study, and work on alternative methods is currently ongoing (e.g. Ruske et al 2018) following new laboratory studies before alternative recommendations can be made.

5f. Issues on clustering. The paper by Crawford et al. (2015) only describe pollen but not if other bioaerosols have been used.

This is correct, the Crawford 2017 paper does not describe laboratory data other than the pollens used. When comparing the fluorescent signature of Cluster 1 and Cluster 3 from Weybourne, in which there is a greater fluorescent signature in channel FL3, the Crawford et al 2017 reference (Page 10, line 9) used is misleading as this infers that there was laboratory data to compare to.

Instead, in the Crawford 2017 paper, Cluster 2 is strongly fluorescent in FL3, similar to Cluster 1 and Cluster 3 from Weybourne (Table 1). However, when considering the size (7.7μm) and shape (Af, 20) of Cluster 2 from Crawford et al 2017, it was speculated that this may represent a bacterial aggregate or a larger dust particle containing uncharacterised bacteria. As per existing bioaerosol studies, cluster profiles were compared to results from Hernandez et al 2016 and Savage et al 2017, which both show a strong FL1 signature for bacteria.

*Table 1 - Comparison between Weybourne Cluster 1 and Cluster 3, and Cluster 2 (Halley) from Crawford et al 2017.*

|  | CL | FL1 | FL2 | FL3 | D(μm) | AF | % Total |
|---|---|---|---|---|---|---|---|
| **Weybourne** | 1 | 3.8 ± 25.2 | 12.5 ± 35.6 | 303.6 ± 295.0 | 5.0 ± 2.2 | 36.6 ± 15.9 | 33.7 |
| **Weybourne** | 3 | 5.1 ± 27.3 | 3.2 ± 15.8 | 192.6 ± 200.8 | 2.0 ± 1.1 | 17.2 ± 8.7 | 50.6 |
| **Halley** | 2 | 135.8 ± 227.4 | 172.1 ± 185.1 | 765.6 ± 535.9 | 7.7 ± 4.0 | 19.9 ± 9.2 | 2.1 |

**To clarify, the Crawford et al 2017 reference has been removed from this sentence. Instead, a new sentence has been added to the manuscript (Section 3.2.1, paragraph 1) illustrating the fluorescent profile of Cluster 2 from the Antarctic site -**

*"…….Laboratory data collected using a WIBS-3 have shown that bacteria such as unwashed E-Coli and Bacillus atrophaeus (BG) spores exhibit higher fluorescence in channel FL3 (Dstl Experiment 2014). The presence of a highly fluorescent FL3 channel was found for Cluster 2 in Crawford et al 2017 which was used to infer a bacterial particle, or dust containing bacteria, as a result of the larger size and shape of the particles in this cluster. This is contrary to other studies which have found a strong FL1 signature for bacteria (Hernandez et al., 2016; Savage et al., 2017)."*

…….Crawford et al. writes that the four pollen types are common in the UK. This is not correct. Two of the four allergens (paper mulberry and ragweed) are rare in the UK. The third in Crawford (birch) is common in the UK, typically with a season in April. This suggest that in this manuscript only Capel Dewi would have had a chance to detect this. The fourth pollen in Crawford et al (2015) is ryegrass……

We note that paper was a limited study. To clarify our results, of the total 18 clusters, following initial comparison of fluorescent signatures from laboratory data to fluorescent channel responses from each cluster, 3 clusters were considered to be pollen fragments (Section 3.2). These were Cluster 1 from Davidstow, Cluster 4 from Weybourne, and Cluster 4 from Chilbolton.

In the manuscript fluorescent signals from three different pollens sampled using a WIBS-3, during the Dstl 2014 experiment, comprising Ryegrass (as in Crawford et al 2017), Aspen, and Poplar pollen. In Crawford et al 2017 'four typical pollens' birch, paper mulberry, ragweed, and ryegrass were used from the sample set.

Two different tree pollens were selected for comparison to the cluster data, these being Aspen pollen, a part of the poplar family, which was selected due to its status as a native tree species to the UK and parts of Europe. Poplar pollen was also selected as was the grass pollen, Ryegrass. This study did not include Paper Mulberry, Ragweed, or Birch as in Crawford et al (2017).

Grass pollens are common from mid-May to July (Met Office 2018) of which the sample period from Chilbolton falls too early (20[th] January to 20[th] March) and the sample period from Weybourne falls too late (17[th] August to 25[th] August). However, at Davidstow, data collection was conducted from 25[th] June to 28[th] August, which is well within the grass pollen season.

Tree pollens, in particular Poplar, are common from around mid-March to early-April (Met Office 2018). The sample period from Weybourne and Davidstow occurs too late in the year for it to be affected by this pollen type. However, at Chilbolton the sample period covers a small part of this period towards the end of the sampling campaign.

Birch pollen was not included in this manuscript, however, as a common UK tree species, it is common from late March to around the middle of May (Met Office 2018). The time period of such would most likely just miss the end of the sampling period for Chilbolton, and start before the sampling period at Davidstow and Weybourne.

….However, the pollen size is typically 30-40, which is above the typical detection limit of the WIBS.

In Appendix A of Crawford et al (2017) the 'four typical pollens' were clustered, and the cluster which accounted for ~70% of the fluorescent material was considered to be representative of the sampled pollens. The average size of this group of these pollen particles (11.8μm) is within the detection range of the WIBS and provided confirmation that ambient Cluster 4 (8.1μm) within the Halley dataset was a potential representative of pollen.

The size of the particles in the manuscript from Cluster 1 in Davidstow (16.5μm), Cluster 4 from Weybourne (4.4μm), and Cluster 4 from Chilbolton (2.4μm) were considered to be pollen fragments owing to the size of these particles being smaller than that of intact pollens. Specifically, pollens such as Birch can range from 24μm to 28μm (Detweiler and Hurst 1930), which would not be detectable by a WIBS. Therefore any clusters with similar spectral signatures which were assumed to be pollen, were considered pollen fragments, as the WIBS would not be able to sample larger sized intact pollen grains, but would be able to detect fragments as described by Savage et al 2017.

Have the authors also calibrated with pollen and have they also used pollen that are less likely to be in the UK atmosphere and less likely to be detected by the WIBS?
As discussed above, the pollens used from the WIBS-3 data (Poplar, Aspen, and Ryegrass) were chosen following research into the different pollen type abundance in the UK, and were selected, as they are most likely to be in the UK atmosphere. These pollens would be likely detected by a WIBS, for example in Savage et al (2017) 14 intact pollens and 13 pollen fragments were sampled and displayed significant fluorescence above the set threshold when using a WIBS-4A.

5g. Issues on clustering. In the paper by Crawford et al (2015) the team has used dry pollen. Dry pollen from commercial samples will have a very different shape to fresh airborne pollen as pollen can take up and loose water. Using dry pollen will generally cause poor calibration of real-time instruments as the shape of dry pollen is very different compared to fresh pollen.

Dry pollen was used within this manuscript (referred to as Dstl Experiment 2014). The authors understand that commercial samples will not necessarily be representative of ambient pollens or other biological material within the atmosphere. Further work comparing the fluorescent signals of ambient and laboratory data of the same particle type is to be conducted in the future.

Secondly has there been any investigations if dry pollen will cause different excitation compared to fresh pollen?

Additional work has been recently conducted (cited as Dstl Experiment 2017 in the manuscript) using dry pollen to compare fluorescence profiles between multiple UV-LIF instruments. As we note in response to reviewer 1, we are planning on conducting a much more thorough evaluation of statistical methods once we have published and had this data appropriately peer reviewed.

6. The methods section are generally good if the issues in section 5 can be solved

We thank the reviewer for this supportive comment and hope that the above has addressed all aforementioned concerns.

7. The citations and reference list seems to be up-to-date with a good selections of citations to new and relevant literature. However the manuscript is not clear where the studies confirms existing knowledge and more importantly where it contributes with new knowledge by positioning the results against published literature

As the reviewer already notes, there are very few studies in this area. This manuscript has aimed to characterise biological particles following HAC analysis, whilst assessing sensitivity to a new recommended threshold, following Savage et al 2017. This paper highlights the method of characterising clusters by firstly comparing the fluorescent profiles to laboratory data, secondly, by assessing the clusters diurnal variation (following grouping as based on their fluorescent profiles), and thirdly, characterising these clusters as based on the response to temperature and relative humidity.

In particular, results were compared to the Hernandez et al 2016 laboratory data study in which different particle types were sampled by a WIBS-4, which was used to help classify the different clusters into groups, with an inferred cluster type. Those in Group 6 were considered to be interferent particles, given the higher FL2 fluorescent profile of these clusters (as found in Hernandez et al 2016). However, it was concluded that Group 6 was not consistent with

interferent particles (as illustrated by Figure 4 in the manuscript) in which of the three clusters one was determined to be bacteria, the other two wet discharged fungal spores.

8. The title of the paper reflects parts of the study, but not the part that try to associate the observed bioaerosols with potential sources (the ArcGIS part)

**To represent the potential sources identification part of the paper, we have changed the title of paper to:**
Characterisation and source identification of biofluorescent aerosol emissions over winter and summer periods in the United Kingdom

9. The abstract cover well the contents of the paper

10. The presentation is generally clear and well-structured but the conclusion might need some work (see point 13)

This was raised by Reviewer 1 and we fully address this issue in response to point 13 below.

11. The language is generally clear and fluent and does not need further improvements

12. The manuscript does not include mathematical formula. However the manuscript describes the use of a third order polynomia with R values between the observations and the polynomia (Table 5). The polynomia is not found anywhere in the manuscript (or in supplementary information) and the results (including low R values) will probably need a discussion.

This is commented on by Reviewer 1 also, and further discussion regarding these results would add to the already lengthy size of the manuscript. As a result, **the statistical analysis section has been removed from the manuscript.**

13. The conclusion is almost two pages and part of the conclusion seems to be a discussion (e.g. the section concerning difficulties in the clustering). Maybe the conclusion should be shortened to make it more sharp and part of the material should be moved to the discussion section.

**The conclusion has been shortened to provide a clearer concluding message (page 23 – 25).**

…If the authors have used calibration of the instrument against known material, then this calibration needs to be described in more detail and in particular how well the instrument is able to identify test samples similar to the calibration material.

**We have added an appendix section with a brief overview of Dstl data collection from 2014 and 2017, and references to published work showing the ability of other UV-LIF spectrometers to identify biological particulates –**

**"*Appendix A: Laboratory characterisation of biological particulates Laboratory data from a WIBS-3D were collected during a series of characterisation studies at the Defence, Science, and Technology Laboratory (Dstl). This data included bacteria comprising unwashed E-Coli and BG spores, and following research into pollen type abundance in the United Kingdom, Poplar, Aspen, and Ryegrass pollen. Additionally, fungal spore data (Alternaria and Cladosporium) collected using a WIBS-4M during an intensive chamber experiment conducted at Dstl, were also used for comparison to the ambient clusters. Further details of this experiment are to be published in 2019. Similar particle types included in this study have been sampled previously by other UV-LIF spectrometers e.g. Hernandez et al 2016, Savage et al 2017. Here, data from both experiments were clustered in the same way as the ambient data (as described in Section 2.3), with the dominant cluster compared to the ambient fluorescent profiles."***

14. There seem to be 60-65 references in the manuscript. This seems appropriate for this type of manuscript

15. There is no supplementary information. The authors might consider if adding supplementary information can improve transparency of the methods and the documentation.

As requested by Reviewer 1, supplementary information has been added to include plots which show the different meteorological variables and the clusters/total fluorescent particles, in addition to other supporting plots.

Specific comments: On page 25, line 4 onwards, the authors write that this is the first time ArcGIS has been used in relation to land cover mapping and bioaerosols to derive emission patterns etc. As far as I know there are many such studies (some of them are in fact in the reference list), but it is the first time it has been done in connection with the WIBS instrument.

This is a miswording and is meant to state that this is the first time using UV-LIF instrumentation in relation to land cover mapping.

This has been changed on page 25 from:

*'To our knowledge this is the first use of both ArcGIS land cover mapping, in association with airborne bioaerosol concentrations, to identify distinctive 5 emission patterns and factors.'*

To:
**'To our knowledge this is the first use of both ArcGIS land cover mapping, in association with airborne bioaerosol concentrations collected using a UV-LIF spectrometer, to identify distinctive emission patterns and factors.'**

**References**

Crawford, I., Robinson, N. H., Flynn, M. J., Foot, V. E., Gallagher, M. W., Huffman, J. A., Stanley, W. R., and Kaye, P. H.: Characterisation of bioaerosol emissions from a Colorado pine forest: results from the BEACHON-RoMBAS experiment, Atmos. Chem. Phys., 14, 8559-8578, https://doi.org/10.5194/acp-14-8559-2014, 2014.

Crawford, I., Gallagher, M. W., Bower, K. N., Choularton, T. W., Flynn, M. J., Ruske, S., Listowski, C., Brough, N., Lachlan-Cope, T., Flemming, Z. L., Foot, V. E., and Stanley, W. R.: Real Time Detection of Airborne Bioparticles in Antarctica, Atmospheric Chemistry and Physics Discussions, pp. 1–21, https://doi.org/10.5194/acp-2017-421, http://www.atmos-chem-phys-discuss.net/acp-2017-421/, 2017.

*Detweiler, HK and Hurst, H (1930): Studies on the pollen content of air, Journal of Allergy, Volume 1, Issue 4, pp.334 -345 (*https://doi.org/10.1016/S0021-8707(30)90229-8)

Hernandez, M., Perring, A. E., McCabe, K., Kok, G., Granger, G., and Baumgardner, D.: Chamber catalogues of optical and fluorescent signatures distinguish bioaerosol classes, Atmospheric Measurement Techniques, 9, 3283–3292, https://doi.org/10.5194/amt-9-3283-2016, 2016.

Met Office 2018 *(*https://www.metoffice.gov.uk/health/public/pollen-forecast*) [accessed 12/10/18]*

Ruske, S., Topping, D. O., Foot, V. E., Morse, A. P., and Gallagher, M. W.: Machine learning for improved data analysis of biological aerosol using the WIBS, Atmos. Meas. Tech. Discuss., https://doi.org/10.5194/amt-2018-126, in review, 2018.

Savage, N., Krentz, C., Könemann, T., Han, T. T., Mainelis, G., Pöhlker, C., and Huffman, J. A.: Systematic Characterization and Fluo- rescence Threshold Strategies for the Wideband Integrated Bioaerosol Sensor (WIBS) Using Size-Resolved Biological and Interfering Particles, Atmospheric Measurement Techniques, pp. 1–41, 2017.